EMBO
Molecular Medicine

# The BACE-1 inhibitor CNP520 for prevention trials in Alzheimer's disease

Ulf Neumann[1],*, Mike Ufer[2],†, Laura H Jacobson[1],‡, Marie-Laure Rouzade-Dominguez[2], Gunilla Huledal[3],§, Carine Kolly[4], Rainer M Lüönd[5], Rainer Machauer[5], Siem J Veenstra[5], Konstanze Hurth[5], Heinrich Rueeger[5], Marina Tintelnot-Blomley[5], Matthias Staufenbiel[1],¶, Derya R Shimshek[1], Ludovic Perrot[1], Wilfried Frieauff[4], Valerie Dubost[4], Hilmar Schiller[3], Barbara Vogg[3], Karen Beltz[3], Alexandre Avrameas[6], Sandrine Kretz[6], Nicole Pezous[2], Jean-Michel Rondeau[7], Nicolau Beckmann[8], Andreas Hartmann[4], Stefan Vormfelde[2], Olivier J David[9], Bruno Galli[9], Rita Ramos[9], Ana Graf[9] & Cristina Lopez Lopez[9],**

## Abstract

The beta-site amyloid precursor protein cleaving enzyme-1 (BACE-1) initiates the generation of amyloid-β (Aβ), and the amyloid cascade leading to amyloid plaque deposition, neurodegeneration, and dementia in Alzheimer's disease (AD). Clinical failures of anti-Aβ therapies in dementia stages suggest that treatment has to start in the early, asymptomatic disease states. The BACE-1 inhibitor CNP520 has a selectivity, pharmacodynamics, and distribution profile suitable for AD prevention studies. CNP520 reduced brain and cerebrospinal fluid (CSF) Aβ in rats and dogs, and Aβ plaque deposition in APP-transgenic mice. Animal toxicology studies of CNP520 demonstrated sufficient safety margins, with no signs of hair depigmentation, retina degeneration, liver toxicity, or cardiovascular effects. In healthy adults ≥ 60 years old, treatment with CNP520 was safe and well tolerated and resulted in robust and dose-dependent Aβ reduction in the cerebrospinal fluid. Thus, long-term, pivotal studies with CNP520 have been initiated in the Generation Program.

**Keywords** Alzheimer's disease; BACE-1 inhibitor; drug discovery; prevention; β-amyloid

**Subject Categories** Neuroscience; Pharmacology & Drug Discovery

See also: **JAD Zakaria & RJ Vassar** (November 2018)

## Introduction

There is high unmet medical need for effective treatment of Alzheimer's disease (AD), one of the most prevalent and debilitating of neurological diseases. More than 6 million people in the United States suffer from dementia, and this number is predicted to rise to 15 million by 2060 (Brookmeyer *et al*, 2018). Currently available pharmacological therapies can treat only the symptoms of AD and have limited benefit. The pathophysiology of AD has been the subject of intense investigation in past decades, and a causal role for aggregated and deposited forms of amyloid-β (Aβ) is supported by a vast body of histopathological, genetic, and biomarker studies (Jonsson *et al*, 2012; Potter *et al*, 2013; Musiek & Holtzman, 2015; Scheltens *et al*, 2016). Consequently, investigational treatments targeting Aβ, such as anti-Aβ antibodies and inhibitors of beta-site amyloid precursor protein cleaving enzyme-1 (BACE-1), are in advanced clinical development. However, several trials in early to moderate stages of dementia have failed to meet their primary endpoints or were stopped at interim analysis (Mullane & Williams, 2013; Hawkes, 2017; Egan

1 Neuroscience, Novartis Institute for BioMedical Research, Basel, Switzerland
2 Translational Medicine, Novartis Institute for BioMedical Research, Basel, Switzerland
3 PK Sciences, Novartis Institute for BioMedical Research, Basel, Switzerland
4 Preclinical Safety, Novartis Institute for BioMedical Research, Basel, Switzerland
5 Global Discovery Chemistry, Novartis Institute for BioMedical Research, Basel, Switzerland
6 Biomarker Discovery, Novartis Institute for BioMedical Research, Basel, Switzerland
7 Chemical Biology and Therapeutics, Novartis Institute for BioMedical Research, Basel, Switzerland
8 Musculoskeletal Diseases, Novartis Institute for BioMedical Research, Basel, Switzerland
9 Global Drug Development, Novartis, Basel, Switzerland
  *Corresponding author. Tel: +41 79 845 6425; E-mail: ulf.neumann@novartis.com
  **Corresponding author. Tel: +41 61 324 0899; E-mail: cristina.lopez_lopez@novartis.com
  †Present address: Idorsia Pharmaceuticals Ltd., Allschwil, Switzerland
  ‡Present address: The Florey Institute of Neuroscience and Mental Health, Melbourne, Vic., Australia
  §Present address: Swedish Orphan Biovitrum AB, Stockholm, Sweden
  ¶Present address: Hertie Institute for Clinical Brain Research, Tübingen, Germany

*et al*, 2018; Honig *et al*, 2018). Recent evidence that Aβ deposition and measurable changes in brain biomarkers occur years before dementia symptoms appear suggests that in these trials, treatment was given too late during the course of the disease to be effective (Villemagne *et al*, 2011; Jack *et al*, 2013; Jones *et al*, 2016). The clinical development of drugs that target Aβ is therefore increasingly focused on treatment during the earlier stages of AD (the newly defined preclinical stage and during the stage of mild cognitive impairment) when the disease-modifying therapy targeting Aβ presumably will be of greatest benefit (Selkoe & Hardy, 2016). If successful, such a preventive treatment could have a profound clinical and public health impact if it is safe, well tolerated, and initiated early enough during the course of the disease to be effective.

BACE-1 is a membrane-bound aspartyl protease required for the processing of the amyloid precursor protein (APP) to form the N-terminus of Aβ peptides and the protein-soluble APPβ (sAPPβ) (Vassar *et al*, 1999). Subsequent intramembrane proteolysis catalyzed by the γ-secretase complex releases amyloid-β peptides of 38–43 amino acids, which form the pathogenic oligomeric and fibrillary Aβ species (Haass, 2004; Masters & Selkoe, 2012). Inhibition of BACE-1 by low-molecular-weight compounds has emerged as a new concept for treatment of AD (Vassar *et al*, 2014; Eketjall *et al*, 2016; Kennedy *et al*, 2016; Timmers *et al*, 2016, 2017; Yan *et al*, 2016; Cebers *et al*, 2017) by preventing the generation and deposition of Aβ rather than just treating the dementia symptoms. However, limited selectivity for BACE-1 over cathepsin D (CatD), the off-target inhibition of which is a principal driver of ocular toxicity, and liver enzyme elevation have led to the termination of some early BACE-1 inhibitors in clinical trials (May *et al*, 2011, 2015; Zuhl *et al*, 2016). The termination of atabecestat (JNJ-54861911) in Phase III trials following liver enzyme elevation was recently announced. Although the small molecule BACE inhibitors, verubecestat and lanabecestat (recently stopped at interim analysis in Phase III trials), have a good selectivity for BACE-1 over CatD, they still exhibit undesirable characteristics: hair depigmentation in animal studies due to high BACE-2 inhibition and strong P-glycoprotein (P-gp)-mediated efflux that limits brain penetration (Cebers *et al*, 2016; Kennedy *et al*, 2016).

We designed CNP520 specifically to avoid the above limitations and thus develop a BACE-1 inhibitor with a safety and tolerability profile suitable for long-term preventive treatment in AD.

We are currently testing CNP520 in the Generation Program (Lopez Lopez *et al*, 2017), a pivotal program designed to assess efficacy and safety in an as yet cognitively unimpaired population at increased risk for developing clinical symptoms of AD based on their age and *APOE4* genotype (NCT02565511 and NCT03131453).

We herein describe the structure and the pharmacological, safety, and early clinical profiles of the BACE-1 inhibitor CNP520, with a special focus on the intended use of this compound for the prevention of AD.

# Results

### CNP520 structural and functional features are distinct from previous BACE-1 inhibitors

We present here the structure of the BACE-1 inhibitor CNP520 for the first time (Fig 1A). It is the result of extensive structural optimization of the 3-amino-1,4-oxazine lead series of BACE-1 inhibitors and a detailed understanding of CNP520's binding to BACE-1 (Fig 1B). Our priorities were to develop a compound (i) with good brain penetration to minimize peripheral exposure and the associated risk of side effects, (ii) with sufficient selectivity over BACE-2 and cathepsins to avoid any clinically relevant interactions with these counter targets, and (iii) without any structural elements in the parent structure or main metabolites that would be mutagenic or genotoxic. Although a detailed description of the drug design process is beyond the scope here, we summarize below the key features of CNP520's structure.

The amino-oxazine head group mediates the key binding interactions to Asp32 and Asp228 in BACE-1 (Fig 1B, Appendix Fig S1, and Appendix Table S1), but, being the most polar part of the molecule, it also largely determines the permeation and distribution properties of CNP520. In a previous series of BACE-1 inhibitors, we discovered that weakly basic compounds show good passive membrane permeation (Lerchner *et al*, 2010). Consequently, the basicity of the cyclic amidine of CNP520 was fine-tuned to pKa = 7.2 by introduction of the electron-withdrawing trifluoromethyl group. Furthermore, we speculated that a bulky group at position 2 of the oxazine ring might reduce binding of CNP520 to P-gp, a major drug transporter located at the blood–brain barrier responsible for drug efflux. Net drug transport into the brain is determined by the balance between passive blood vessel membrane permeation and P-pg-mediated outward transport (Meredith *et al*, 2008). *In vitro* investigation of CNP520 transport properties in Madin–Darby canine kidney (MDCK) cells expressing human P-gp (Rueeger *et al*, 2011) proved our hypothesis: The rate for apical-to-basolateral transport of $5.6 \times 10^{-6}$ cm/s indicated good passive membrane permeation, whereas the P-gp-driven basolateral-to-apical transport rate of $14.3 \times 10^{-6}$ cm/s was not substantially higher. In agreement with the *in vitro* data, a distribution study in rats showed comparable unbound CNP520 concentrations in the brain and in blood (Fig 1C), confirming that significant efflux did not occur.

Enzyme inhibition assays showed CNP520 to be a potent BACE-1 inhibitor that is selective for BACE-1 over other human pepsin-like aspartic proteases, including BACE-2 and CatD (Table 1). BACE-1 shares high structural similarity with the catalytic and ligand binding sites of these enzymes; however, we increased selectivity for BACE-1 by exploiting the size differences of remote substrate binding pockets, in particular the extended S3 pocket (Schechter & Berger, 1967). We discovered that a picolinic acid with *para*-trifluoromethyl substitution, pointing into the extended S3 pocket, provided threefold selectivity for BACE-1 over BACE-2 and 20,000-fold selectivity for BACE-1 over CatD, which translated to functional selectivity *in vivo* (see below). Finally, we identified the central fluoropyridyl moiety as a non-genotoxic building block that could replace earlier aniline-type substructures that generated aniline metabolites upon hydrolysis of the amide bond (Neumann *et al*, 2015). Thus, *in vivo* metabolism of CNP520 forms an *ortho*-aminopyridine-containing metabolite with no known safety flag (Fig 1D) and does not generate aniline with its associated genotoxicity risk (Smith, 2011).

CNP520 is a moderately lipophilic compound; the logD (partition coefficient octanol/buffer pH 6.8) of 3.5 is in the target range for an orally available and brain-penetrating drug. However, the non-specific binding of CNP520 to plasma proteins is relatively high (ranging from 95.9% in human to 98.9% in mouse); therefore, the unbound concentration of CNP520 in plasma is low, typically below 0.1 μM at pharmacologically active doses in animals and humans (see below).

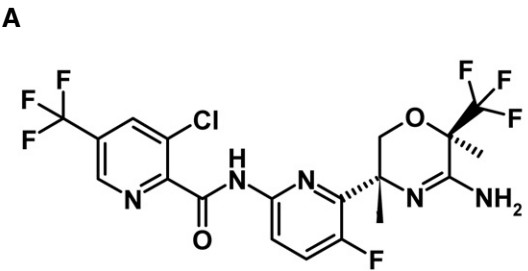

**A**

MW: 513.8
logD (pH 6.8): 3.5
pK$_a$: 7.2

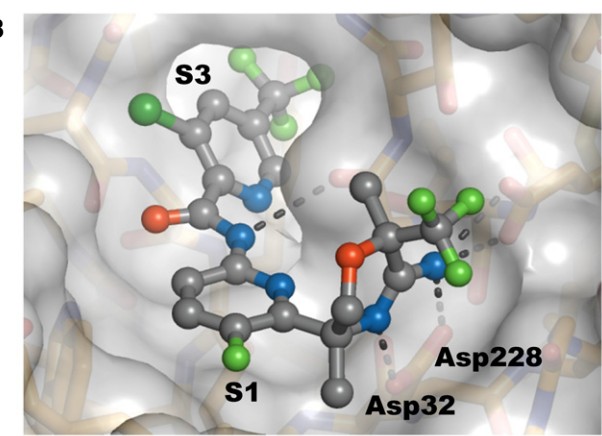

**B**

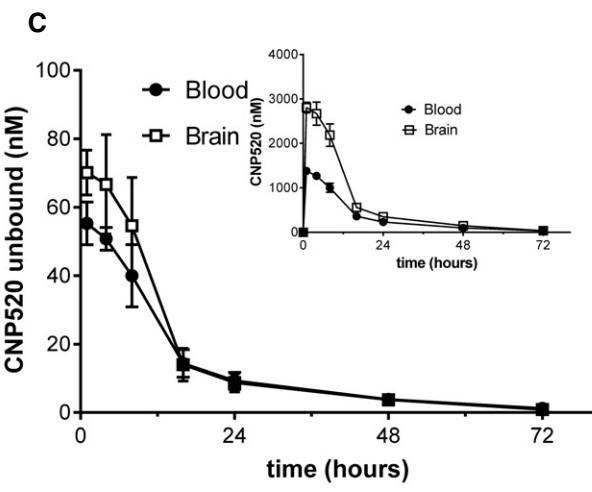

**C**

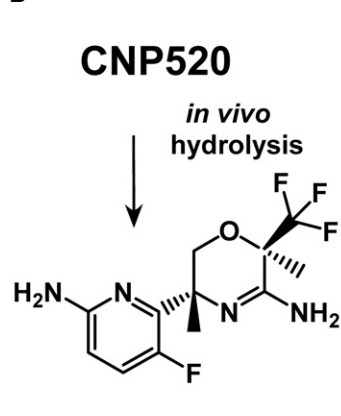

**D**

CNP520

*in vivo* hydrolysis

**Figure 1.  Design of CNP520.**

A  Molecular structure.

B  X-ray structure of CNP520 bound to the BACE-1 active site (resolution: 1.35 Å), with binding pockets S1 (filled by P1) and S3 (filled by P3) and catalytic residues Asp32 and Asp228.

C  Blood and brain tissue levels of CNP520 in the rat after a 30 μmol/kg (15.4 mg/kg) oral dose. Shown are unbound fraction (main graph) and total levels (insert) of CNP520 in the blood and brain, using values for rat plasma protein binding of 97.8% and unspecific binding to brain homogenate of 99%. Values are mean ± SD, with $n$ = 5 per time point.

D  Structure of the aminopyridine generated by metabolic cleavage of the amide bond in CNP520.

## CNP520 has a benign non-clinical safety profile

During our evaluation of the non-clinical safety of CNP520, we paid special attention to the physiological effects that have been ascribed to BACE-1 knockout (KO) mice, including demyelination/thinning of myelin sheets, muscle spindle and proprioception changes due to decreased processing of neuregulin-1, and retina atrophy via impairment of vascular endothelial growth factor-1 cleavage-induced choroidal neovascularization (Willem *et al*, 2006; Cai *et al*, 2012; Cheret *et al*, 2013). None of the effects listed above were observed in toxicology studies with CNP520.

In long-term toxicology studies, dose-limiting non-specific central nervous system (CNS) effects (including impaired mobility and tremor) occurred in dogs at ≥ 30 mg/kg/day and in male rats (adverse effects only observed at doses ≥ 500 mg/kg/day). In female rats, we found delayed ovulation and slightly reduced fecundity and fertility indices (74 versus 90% in control animals) at ≥ 30 mg/kg/day, and focal skeletal muscle atrophy in gastrocnemius and quadriceps muscles without functional effects at 200 mg/kg/day. After CNP520 withdrawal, all changes either trended toward being, or were, fully reversible. The organs that were affected by drug toxicity in these animal studies either were not relevant to the clinical target population or the toxicities occurred at a dose substantially higher than the efficacious clinical dose. No morphological changes were identified in the CNS (brain, spinal cord, and central nerves), peripheral nervous system (peripheral nerves, spindle cells, and neuromuscular junctions), liver, endocrine pancreas, retina, or the skin of any species.

**Table 1. *In vitro* potency of CNP520.**

| Enzyme | IC$_{50}$ $\pm$ SEM (nM) |
|---|---|
| Human BACE-1 | 11 $\pm$ 0.4 |
| Mouse BACE-1 | 10 $\pm$ 0.3 |
| Human BACE-2 | 30 $\pm$ 1.0 |
| Human cathepsin D | 205,000 $\pm$ 28,200 |
| Human cathepsin E | 66,400 $\pm$ 13,000 |
| Porcine pepsin | > 250,000 |
| CHO cells (wild-type APP) | 2.8 $\pm$ 0.2 |
| CHO cells ("Swedish" APP) | 44 $\pm$ 0.4 |

Recombinant catalytic domains were used for the human aspartyl proteases and for mouse BACE-1. IC$_{50}$ values are means of three individual CNP520 batches, each measured in duplicate. IC$_{50}$ values for the inhibition of Aβ40 release from APP-transfected CHO cells are means from six different CNP520 batches (each in triplicate, APP-wild-type cells) or three different batches (triplicates, APP-Swedish cells).

We investigated the safety pharmacology profile of CNP520 *in vitro* and *in vivo*. In our assessment of CNP520 cardiovascular safety, we paid special attention to the human Ether-à-go-go-Related Gene (hERG) potassium channel in the heart, as it is an important potential off-target for aromatic compounds with a basic/amphiphilic structure (Kalyaanamoorthy & Barakat, 2017). Due to their basic cyclic amidine, combined with a rather hydrophobic P3 substituent, BACE-1 inhibitors are potential hERG ligands. *In vitro*, CNP520 inhibited hERG with an IC$_{50}$ of 3.2 μM, 67-fold above the free $C_{max}$ of the highest clinical dose tested in a 3-month human study (0.048 μM at 85 mg). We observed no effect on heart rate, ECG morphology, or blood pressure in dogs at doses up to 200 mg/kg (free plasma $C_{max}$ = 0.23 μM) or in the repeat dose toxicology studies (2, 4, 13, and 39 weeks' duration).

To further delineate the safety pharmacology profile of CNP520, we investigated its affinity for a panel of neurotransmitter receptors associated with drug abuse or dependences. Although a number of the receptors had binding affinities in the low micromolar range, the CNP520 concentrations used in these experiments were higher than the free drug concentration obtained in human plasma at the clinically relevant dose of 85 mg/day (Appendix Table S2). Thus, the safety margins are robust, and the potential for functional interactions between CNP520 and all of the receptors tested is low. In agreement with these results, neither rats nor dogs displayed clinical signs indicative of drug withdrawal (i.e., rearing, or altered locomotor activity, body temperature, food consumption, or body weight) after cessation of repeated dosing. Therefore, no potential for drug dependencies was identified from *in vitro* or *in vivo* toxicological studies.

**CNP520 has no relevant side effects due to inhibition of BACE-2 or cathepsin D**

CNP520 is highly BACE-1 specific, with the exception of BACE-2 (Table 1 and Appendix Table S2). To investigate whether CNP520 has any effects on melanin synthesis and hair pigmentation, we dosed C57/BL6 mice with CNP520 for 8 weeks; NB-360, which inhibits BACE-1 and BACE-2 equipotently and leads to hair depigmentation in mice because of reduced PMEL-17 processing (Shimshek *et al*, 2016), was used as positive control. No change in hair color was observed with CNP520 at doses that provided more than 90% Aβ40 reduction in the brain (Fig 2A). NB-360 in contrast caused obvious hair depigmentation in all mice, starting at 2–3 weeks. Neither did we observe any changes in other pigmented organs (hair follicle, choroid, and retina pigmented epithelium evaluated by histopathology).

The complete absence of hair depigmentation in the CNP520 group was surprising because the threefold selectivity for BACE-1 over BACE-2 was not expected to be sufficient to cause such a striking difference. We speculated that both the concentration of CNP520 or NB-360 in the tissue in combination with their respective potencies for BACE-2 would also play a role in determining the level of inhibition of BACE-2 and thus whether depigmentation would occur. Therefore, we determined CNP520 and NB-360 total concentrations in the skin of treated mice and calculated the ratios between these concentrations, and the concentrations known to achieve 50% inhibition (IC$_{50}$) of BACE-2 (Fig 2B). This ratio was more than 10-fold lower for CNP520 than for NB-360, indicating that CNP520 is less available to inhibit BACE-2 than is NB-360. We then estimated the biologically more relevant ratio between the free skin concentration and the IC$_{50}$ for BACE-2, assuming a free fraction of 1% in skin for both CNP520 and NB-360. The free CNP520/BACE-2 IC$_{50}$ ratio was in the range of 1.6–4.6, suggesting that BACE-2 in skin may only be partially inhibited by therapeutic concentrations of CNP520, because the available drug concentration is of the same order of magnitude as the IC$_{50}$ for BACE-2. In contrast, for NB-360, free skin concentrations were estimated to be 17-fold to 114-fold higher than the BACE-2 IC$_{50}$, pointing to a potentially high level of *in vivo* inhibition. We conclude that the combination of threefold selectivity for BACE-1 over BACE-2 and the comparatively low distribution to the skin may explain the absence of CNP520-induced depigmentation in this mouse study. In long-term CNP520 animal toxicology studies, including 9 months dosing in dogs, we found no signs of depigmentation in any study.

In agreement with the 20,000-fold selectivity of CNP520 over CatD, CNP520 showed no retinal degeneration in our long-term toxicological studies in rats and dogs.

**CNP520 does not induce cerebral micro-hemorrhages**

In preclinical models, amyloid-based immunotherapies (particularly monoclonal antibodies) have been associated with cerebral micro-hemorrhages (CMH), the equivalent of amyloid-related imaging abnormalities (ARIA) in humans. We examined the occurrence of CMH following treatment with CNP520 in APP23 mice, a transgenic strain that develops vascular Aβ pathology and displays an increased incidence of age-related CMH after treatment with anti-Aβ antibodies (Beckmann *et al*, 2011). To identify potential risk for CMH, 18-month-old APP23 mice were treated with CNP520 (55 mg/kg/day) for 3 months. CNP520 did not increase CMH frequency or severity relative to vehicle-treated animals as assessed by in-life magnetic resonance imaging (Fig EV1A) and histopathological investigations including staining to identify hemosiderin (Fig EV1B), unlike the positive control antibody β1 (Beckmann *et al*, 2011).

**Single CNP520 doses induce long-lasting Aβ reduction in animals**

A rat distribution study with 15.4 mg/kg CNP520 (Fig 1C) showed a 50 nM peak concentration of unbound CNP520 in the brain, which was fivefold higher than the IC$_{50}$ for BACE-1 inhibition *in vitro*. This

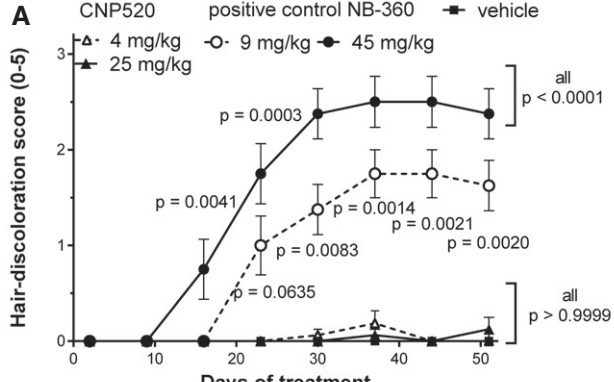

**B**

| | Dose mg/kg | % Aβ40 in brain relative to vehicle alone | Total skin concentration / BACE-2 IC$_{50}$ |
|---|---|---|---|
| CNP520 | 4 | 55% | 160 |
| | 25 | 8% | 460 |
| NB-360 (control) | 9 | 66% | 1700 |
| | 45 | 32% | 11400 |

**Figure 2. CNP520 treatment of male C57/BL6 mice for 8 weeks did not cause hair depigmentation.**

A Subjective assessment of hair depigmentation (score 0–5, with 0 = completely black and 5 = completely white) at two doses of either CNP520 or the BACE-1/-2 inhibitor NB-360 (Shimshek et al, 2016; mean ± SEM, n = 8/group), statistical comparisons were made for every scoring day, versus vehicle using Kruskal–Wallis with Dunn's *post hoc* test.

B Summary of doses, steady-state effects on Aβ40 in brain (% compared to vehicle-treated mice) and the ratios between total skin concentrations of CNP520 and NB-360 and their respective IC$_{50}$ values for inhibition of BACE-2 *in vitro*.

suggested that CNP520 might reduce Aβ in the rat brain after a single oral dose. Therefore, we measured Aβ40 in rat brain tissue 4 h after oral administration over a range of doses. Effects on Aβ40 were dose-dependent: We observed an 89.3 ± 4.5% (mean ± SD) reduction compared to untreated controls at the highest dose (Fig 3A). The oral dose required for 50% lowering of rat brain Aβ40 (ED$_{50}$) was 2.4 ± 0.31 mg/kg. CNP520 showed a long duration of action in the rat, as indicated by ~50% Aβ40 reduction 24 h after a single oral 30 μmol/kg (15.4 mg/kg) dose, in both rat brain and CSF (Fig 3B). In addition, the effect-over-time curves for brain and CSF almost completely superimposed, showing that Aβ pharmacodynamics in CSF mirrors very well that in brain tissue. To follow this up, we chose dog as a non-rodent model, since dog BACE-1 protein shows a high sequence identity to human BACE-1 (82.4%), and effects of BACE-1 inhibitors on Aβ in dog CSF have been investigated (Mattsson et al, 2012). We collected CSF from 3-month-old beagle dogs, via an implanted ventricular port over 7 days after oral administration of 3.1 mg/kg (6 μmol/kg, single dose) CNP520. We detected CNP520 in blood for 192 h and in CSF for 72 h (Fig 3C). We determined the non-specific binding of CNP520 in dog plasma (97.5%) and calculated the free fraction of CNP520 in dog blood. The ratio of unbound CNP520 in dog CSF to unbound CNP520 in blood was 0.7, showing that CNP520 is efficiently distributed to the central compartment. Both Aβ40 and Aβ42 concentrations in CSF showed a > 75% reduction at 12–48 h after dosing and returned slowly to baseline over the next 7 days (Fig 3D). Pharmacokinetic/pharmacodynamic (PK/PD) modeling provided estimates for the IC$_{50}$ *in vivo* of 103.1 ± 12.3 nM for total CNP520 and 3.3 ± 0.03 nM for unbound CNP520 in dog blood.

**CNP520 reduces Aβ load and neuroinflammation in APP-transgenic mice**

We were interested to investigate the effects of CNP520 in a disease model that more closely resembles the situation in an AD brain, where Aβ deposition develops over time. For this, 12- to 14-month-old APP23-transgenic mice, which at this age display Aβ deposition

and plaque pathology in the cortex (Sturchler-Pierrat & Staufenbiel, 2000), were given CNP520 over a 6-month period. CNP520 was delivered in food at a concentration equivalent to a daily oral dose of 4 mg/kg (low dose, 0.03 g/kg food) or 40 mg/kg/day (high dose, 0.3 g/kg food). CNP520 concentration in blood and brain was approximately dose-proportional at the end of the study (Fig 4A). We used immunohistochemistry with an antibody, NT12, which recognizes both Aβ40 and Aβ42 to examine the effect of CNP520 on Aβ deposition in this mouse strain. The total Aβ-plaque area (normalized to sample area) in the cortex significantly increased in the vehicle-treated group compared with baseline and decreased in CNP520-treated mice in a dose-dependent manner compared with vehicle (Fig 4B). The total amount of Aβ40 and Aβ42 in formic acid-solubilized forebrain was determined by Aβ immunoassay. Compared with vehicle-treated mice, the high dose of CNP520 profoundly reduced the brain load of deposited Aβ40 and Aβ42, keeping it close to baseline levels (Fig 4C and D). We also analyzed the levels of soluble APP metabolites in mouse brains: Treatment with CNP520 decreased the direct BACE-1 product sAPPβ in a dose-dependent manner, whereas soluble APPα (sAPPα), the product of the non-amyloidogenic pathway of APP metabolism, increased (Appendix Fig S2), showing that CNP520 treatment shifts APP metabolism away from the amyloidogenic pathway. Next, we investigated the effects of long-term Aβ reduction on downstream pathology relevant for Alzheimer's disease, in particular neuroinflammation. Specifically, we used double-immunofluorescence staining with NT12+anti-Iba1 and NT12+anti-GFAP to examine the number and localization of microglia and activated astrocytes, respectively (Fig 4E). In non-treated APP23 mice, these inflammation markers increased approximately threefold compared to baseline: CNP520 prevented this increase in a dose-dependent manner for astrocytes stained with anti-GFAP, but not for Iba1-stained microglia (Fig 4F and G). Because the Aβ plaque surface is the major site of glia cell activation, we separately analyzed the responses of plaque-associated and non-plaque-associated populations of microglia and astrocytes to CNP520 treatment.

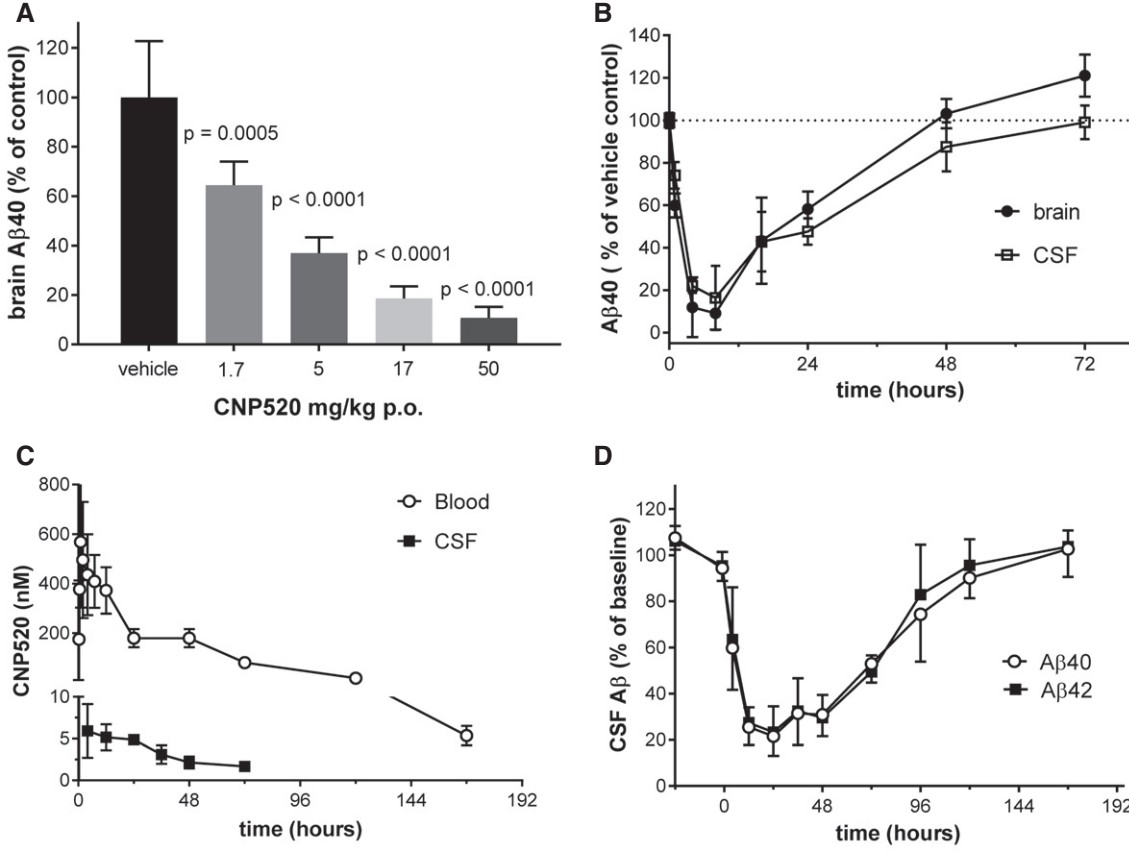

**Figure 3.  Oral administration of CNP520 to rats and dogs.**

A  Doses from 1.5 mg/kg (3 μmol/kg) to 51.3 mg/kg (100 μmol/kg) were given by oral gavage to rats (*n* = 5 per dose); animals were killed 4 h later for determination of Aβ40 in brain tissue. Values are mean ± SEM as determined by ANOVA (GraphPad Prism).

B  Time-dependent reduction of Aβ40 in rat brain and CSF after a single 15.4 mg/kg (30 μmol/kg) oral dose of CNP520 (mean ± SD, *n* = 5).

C  CNP520 levels in dog blood and CSF, following a 3.1 mg/kg (6 μmol/kg) oral dose (mean ± SD, *n* = 4).

D  Time-dependent reduction of Aβ40 and 42 in dog CSF, after a 3.1 mg/kg oral CNP520 dose (mean ± SD, *n* = 4).

Plaque-associated microglia displayed a CNP520 dose-dependent reduction, whereas microglia distant from plaques did not (Fig EV2A and B). The plaque-associated Iba1-positive area (normalized to total sample area) showed a strong correlation with the plaque area, but the correlation was very weak for the non-plaque-associated Iba1 area (Fig EV2C and D), indicating that these mice display a more generalized microglia activation in brain regions distant from plaques. In absolute cell numbers, this CNP520-insensitive microglia population outweighed the relatively small number of

plaque-associated and CNP520-sensitive microglia, giving rise to the overall weak and non-significant treatment effect on the total microglia population. In contrast, CNP520 reduced the number of activated astrocytes both in the plaque-associated and in the non-plaque-associated populations (Fig EV2E and F), and both normalized plaque-associated and non-plaque-associated GFAP area correlated with plaque area (Fig EV2G and H). The reason for the different sensitivity of microglia and astrocytes distant to plaques is currently not clear. In conclusion, chronic CNP520 prevented the

**Figure 4.  Chronic therapeutic treatment of APP23 mice with CNP520.**

Male APP23 mice (age at baseline: 12–14 months old) were dosed with CNP520 in food pellets for 6 months: Low dose 0.03 g CNP520/kg food, corresponding to a daily oral dose of 4 mg/kg; high dose 0.3 g/kg food, corresponding to a daily oral dose of 40 mg/kg.

A  CNP520 levels in blood and cerebellum.

B  Total cortical plaque area (normalized to total sample area) (*10⁻²), stained with antibody NT-12.

C  Formic acid-soluble Aβ40 in forebrain.

D  Formic acid-soluble Aβ42 in forebrain.

E  Double-immunofluorescence-stained brain sections with NT12 (against amyloid-β, green) and Iba1 (against microglia, red) or GFAP (astrocytes, red) for the different treatment groups as indicated. Scale bar: 500 μm.

F  Quantification of total Iba1-positive microglia (normalized to total area) (*10⁻²), only a subset of samples analyzed.

G  Quantification of total GFAP-positive astrocyte area (normalized to total area) (*10⁻²).

Data information: (A, B, C, D, and G) Baseline, *n* = 10, vehicle, *n* = 17, CNP520 low dose, *n* = 14, CNP520 high dose, *n* = 13. (F) All groups *n* = 7. Date are mean ± SEM. Data were analyzed with one way ANOVA and Dunnett's multiple comparison test.

     

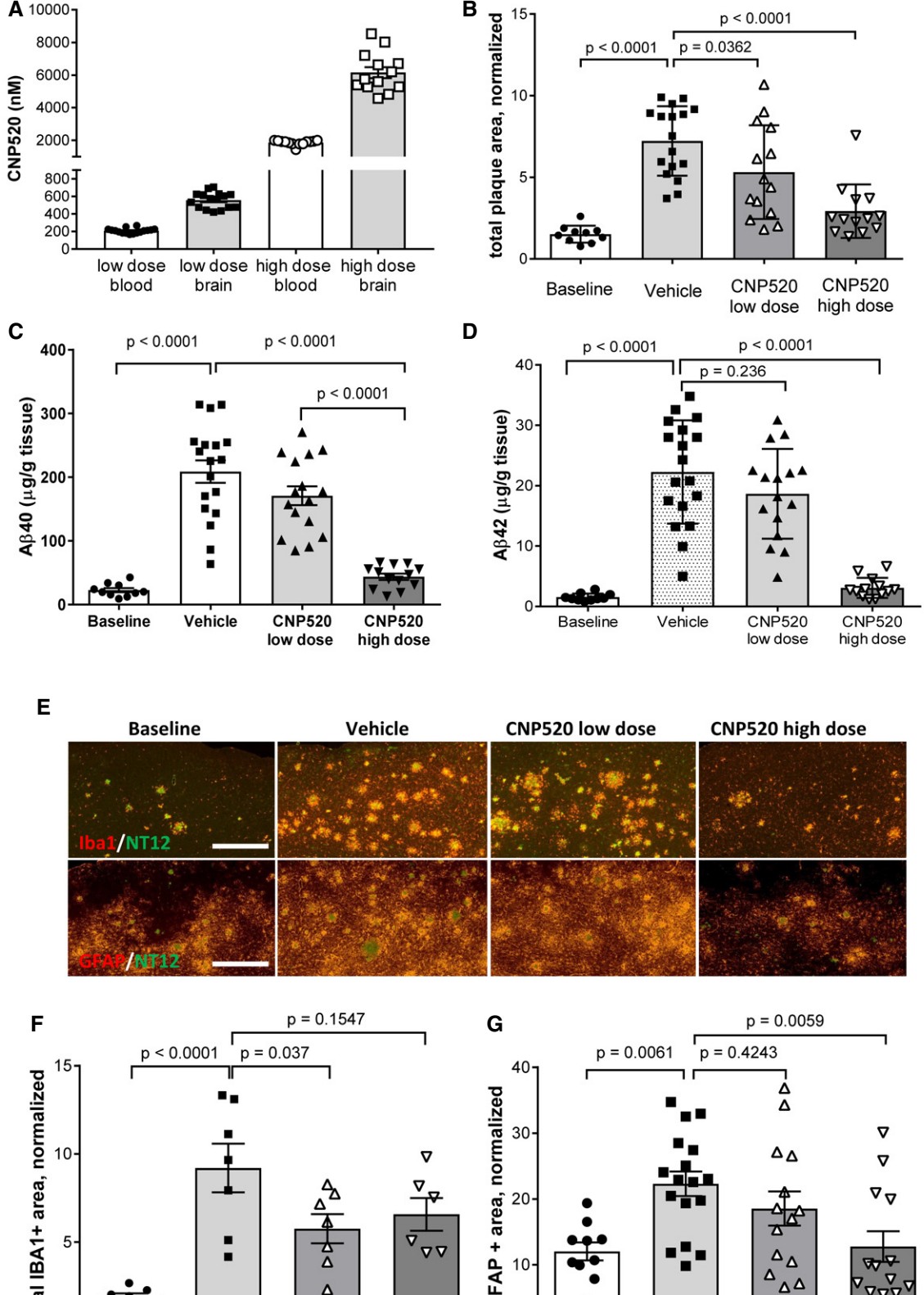

Figure 4.

## CNP520 non-clinical safety profile translates into early clinical studies

In early clinical studies, we generated data that support the safety and tolerability of CNP520 in humans. To date, we completed five human studies with 335 healthy participants exposed to CNP520 (Appendix Tables S3 and S4), including a Phase IIa study of 3-month duration. Only the single ascending dose studies were performed in adults 18–55 or 20–45 years of age, whereas the participants in all subsequent single and multiple dose studies were cognitively normal male and post-menopausal female subjects ≥ 60 years of age, similar to the age range of our target population in the Generation Program.

In participants ≥ 60 years of age, we observed no dose-limiting safety or tolerability findings up to 750-mg single dose, 300 mg for 2 weeks (maximum multiple dose tested), and up to 85 mg for 3 months (Appendix Tables S5–S8). Across completed studies, the adverse event (AE) incidence was similar for CNP520 and placebo in subjects ≥ 60 years, with the exception an imbalance of skin-related AEs observed in the 3-month study, mostly driven by pruritus (18% for CNP520 versus 4% for placebo). Each of these events was mild and transient, except for a single case of spontaneously reported generalized pruritus of moderate severity (not associated with rash) leading to discontinuation of CNP520 at 85 mg. Scheduled dermatological monitoring performed during the study revealed no drug-related clinically relevant findings (pruritus, hypopigmentation, or rash). There were no clinically relevant alterations of laboratory findings, vital signs, electrocardiography data, or any indication for systematic changes over time or as a function of dose. No alterations were found during the ophthalmological evaluation (assessed by visual acuity and visual field).

We detected no indication of impaired neurological function, based on routine neurological examination and cognitive testing (as assessed by the Cogstate Battery). One subject on 85 mg discontinued due to an AE of transient global amnesia (moderate), suspected to be drug-related; however, according to the neurologist's examination, it was possibly due to transient ischemic attacks. No other neurological symptoms were associated.

In concordance with the lack of potential for drug dependencies identified from *in vitro* and *in vivo* toxicological studies, we identified no signal for abuse potential, based on the low overall incidence of abuse-related AEs without dose dependency. The incidence of post-treatment cessation AEs was higher with placebo than on CNP520, suggesting that there were no withdrawal effects.

## CNP520 shows dose-proportional exposure in plasma and CSF

In human pharmacokinetic studies, CNP520 displayed a moderate absorption rate ($T_{max}$ within 1–8 h after dose), and mean terminal $T_{1/2}$ was 61.3–83.8 h in healthy adult participants and 81.4–109 h in participants ≥ 60 years of age following single dose administration (Fig 5A, Appendix Table S9). This long half-life indicated that CNP520 is suitable for once-daily dosing in humans. CNP520 plasma exposure ($C_{max}$ and AUC) increased approximately in proportion to dose following both single and repeated dose administration. Upon daily dosing, CNP520 plasma exposure increased within the first

month of administration in participants ≥ 60 years, but remained stable for the two additional months of dosing (Fig 5B). CNP520 steady-state plasma levels ranged from 11.9 ng/ml (2-mg dose) to 425 ng/ml (85-mg dose), corresponding to 23 and 828 nM, respectively. Intake of a high-fat breakfast increased $C_{max}$ and $AUC_{0–72\ h}$ only slightly, suggesting no significant food effect. There was no major difference in exposure between healthy adult participants and healthy participants ≥ 60 years (Appendix Table S10).

CNP520 distributed to CSF in a dose-proportional fashion (Fig 5C and D), and mean $C_{max}$ was 20.4 ng/ml (39.7 nM) following a single dose of 750 mg. The mean ratio between CSF and plasma concentrations was 0.02–0.03, which is similar to the fraction of unbound CNP520 in plasma (Appendix Table S11). Inter-subject variability (coefficient of variation) in both plasma and CSF PK parameters was approximately 20–35%.

## CNP520 robustly reduces human CSF Aβ

In humans, determination of the direct BACE-1 product sAPPβ and of Aβ peptides in CSF allows measurement of the effect of CNP520 on APP processing in the brain (Blennow *et al*, 2010). We investigated CSF Aβ and sAPPβ levels in humans by serial CSF sampling via an intrathecal catheter. Single CNP520 doses (10, 90, 300 or 750 mg) were administered to adults ≥ 60 years of age, and CSF was sampled at six time points starting at 2 h pre-dose (baseline) up to 34 h. A dose- and time-dependent reduction in CSF Aβ40 (Fig 6A) occurred, with a maximum 79.1 ± 8.9% (mean ± SD) reduction from baseline with the 750-mg dose. CNP520 treatment also led to a dose-dependent decrease in sAPPβ (Fig 6B) and an increase in sAPPα, the product of the competing, non-amyloidogenic APP-cleavage pathway (Fig 6C). At the two highest doses, 300 mg and 750 mg, the response curves for Aβ40 and sAPPβ were superimposable, suggesting that CNP520 had inhibited BACE-1 to saturation (Fig 6A and B). Under these conditions, where Aβ production has completely stopped, the curves most likely reflect the rate of physiological clearance of Aβ40 and sAPPβ from human CSF. The rate constant for clearance of Aβ40 from human CSF was 0.068 ± 0.002/h, corresponding to a half-life of 10.2 h (Appendix Fig S3), close to previous estimations using a kinetic isotope labeling technique (Bateman *et al*, 2006; Patterson *et al*, 2015). Clearance of sAPPβ was slower, with a rate constant of 0.043 ± 0.004/h ($t_{1/2}$ = 16.2 h).

The results from the single dose study allowed the determination of CNP520 doses for a 2-week, once-daily multiple dosing study (10, 30, 90, and 300 mg). CSF was sampled by single lumbar punctures at baseline and 24 h after the last dose, and the change in Aβ concentrations from baseline to end of treatment was determined. Again, the Aβ reduction in participants exposed to CNP520 was dose-dependent, ranging from 59.8 ± 9.88% at a 10 mg daily dose up to a maximum of 93.9 ± 2.09% at a 300 mg daily dose (mean ± SEM; Fig 6D).

After we had established the PK/PD relationship for CNP520 in humans, we selected a range of doses from 2 to 85 mg for a Phase IIa study of 3-month duration in healthy participants ≥ 60 years of age, with *n* = 25 per dose group. All study participants underwent CSF sampling at baseline and at the end of treatment; in addition, we collected CSF at a number of intermediate time points (*n* = 2–7/time point). Overall, the magnitude and time-course of the PD effects were consistent with predictions from Phase I data. After an initial 2–4 weeks of treatment, Aβ40 lowering remained stable for the rest of

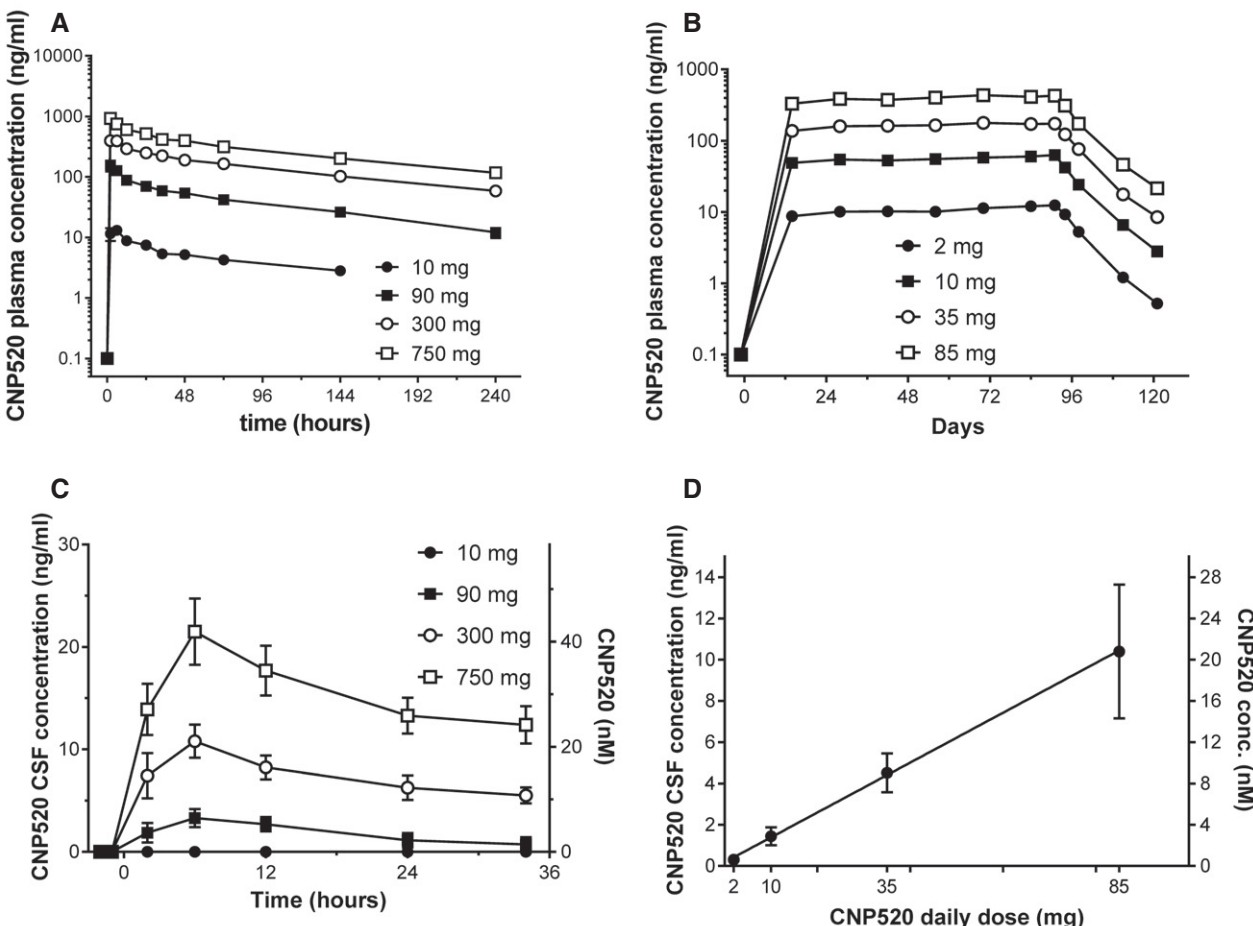

**Figure 5.  Pharmacokinetics of CNP520 in adults ≥ 60 years of age.**

A   Mean plasma concentration versus time profiles after single dose.
B   Mean plasma concentration profiles for 91 days of dosing and post-dose concentrations up to day 121.
C   Time versus CSF concentration profiles after single dose of CNP520 (mean ± SEM, n = 6–8).
D   Dose–CSF concentration profiles after 3 months of dosing (mean ± SEM, n = 22–24).

the 3-month treatment period in all dose groups (Fig 7A). At the end of treatment, significant reduction in CSF Aβ40 concentrations versus placebo was observed at all doses, ranging from 22.6 ± 2.1% at a 2-mg daily dose to 90.7 ± 0.37% at a 85-mg daily dose (mean ± SEM, $P < 0.0001$, n = 21–25; Fig 7B). Very similar effects were observed for responses in CSF Aβ38, Aβ40, and Aβ42 (Appendix Fig S4). Since we covered the full effect range with the doses tested, we were able to determine the $ED_{50}$ relating total plasma CNP520 concentration to inhibition of CSF Aβ40. Fifty percent reduction was reached at 37.5 ± 2.9 ng/ml (73.5 ± 5.8 nM) CNP520 in plasma. When corrected for plasma protein binding, $ED_{50(free)}$ was 3.3 nM, which is practically identical to the $IC_{50}$ observed *in vitro* for the inhibition of Aβ release from cells (Table 1). No apparent differences were observed between genders in terms of PK or PD.

**APOE4 genotype and baseline brain amyloid-β do not influence the CSF Aβ reduction**

Because the Generation Program enrolls people at risk of developing AD based on age and genotype (presence of the *APOE4* gene), we

investigated the ability of CNP520 to inhibit Aβ40 generation preclinically in the *APOE4*-TR mouse model, in which the mouse *Apoe* gene is replaced by the human *APOE4* gene (Castellano *et al*, 2011). We observed a dose-dependent reduction in Aβ40, with 77% reduction at the highest dose of 17 mg/kg (30 μmol/kg), similar to that observed in normal rats (Figs 3 and 8A), suggesting a robust response to CNP520 in an *APOE4* genetic background.

To compare CNP520-induced lowering of human CSF Aβ40 in the presence or absence of the *APOE4* genotype, we analyzed the occurrence of the *APOE4* genotype in the Phase IIa study, which was available for 108 out of 125 participants. Seventy-five participants did not carry the *APOE4* gene, 32 participants carried one copy, and one participant carried two copies. This analysis was done in retrospect, and the presence of the *APOE4* allele was not considered when randomizing participants to the dose groups.

In addition to the *APOE4* genotype, we also determined the CSF Aβ42/40 ratio at baseline, because this ratio provides information on brain Aβ deposition (Fig 8B). Previous studies in patients with mild cognitive complaints suggest that a ratio < 0.09 is indicative of

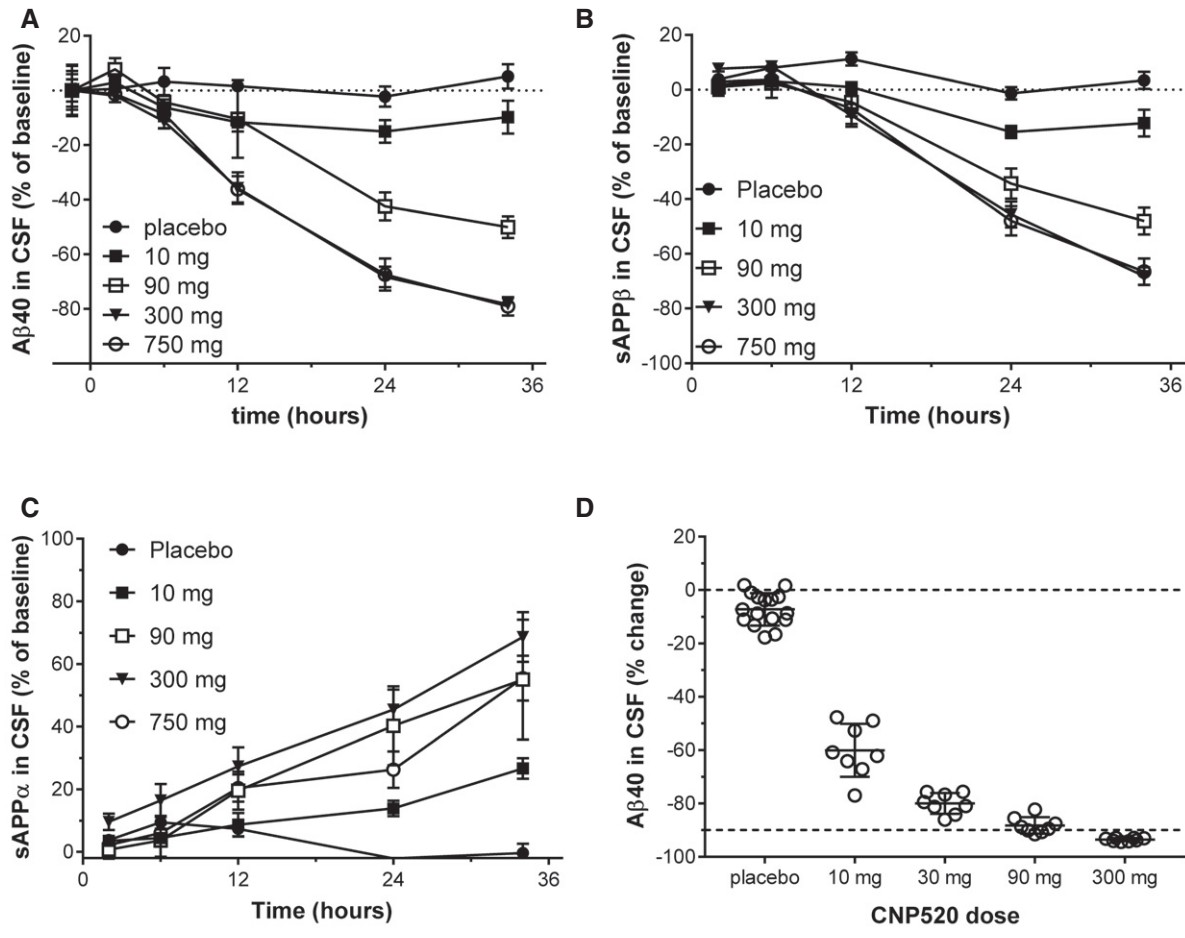

**Figure 6.  Effects of CNP520 administration on major APP metabolites in the CSF of healthy study participants ≥ 60 years of age.**

A–C    CSF was sampled via a lumbar catheter at time points from 2 h pre- to 34 h post-administration of a single dose of CNP520 (mean ± SEM, *n* = 6–8). (A) Changes in Aβ40 relative to baseline values. (B) Changes in sAPPβ relative to baseline values. (C) Changes in sAPPα relative to baseline values.

D    Dose–response of Aβ40 to CNP520 relative to baseline Aβ40 levels after 2 weeks of dosing. CNP520 was administered daily for 14 days; CSF was sampled via a lumbar catheter 2 h before dosing and 24 h after the last dose (mean ± SD, *n* = 8 in treatment groups, *n* = 16 in placebo group).

elevated brain Aβ (Janelidze *et al*, 2016; Pannee *et al*, 2016). A higher number (33%) of *APOE4* carriers presented with Aβ42/40 ratio < 0.09, compared to 15% of non-carriers. The effects of CNP520 treatment on the amount of Aβ40 in CSF were very similar in *APOE4* carriers and non-carriers (Fig 8C). Furthermore, the effects of CNP520 treatment on Aβ40 in CSF were comparable in participants with normal (≥ 0.09) or with reduced (< 0.09) Aβ42/40 ratio in CSF at baseline (Fig 8D). We investigated whether a 3-month CNP520 treatment would affect the Aβ42/40 ratio in CSF by comparing this ratio at baseline with that at day 91. When all subjects were included in the analysis, no change was observed, mainly because subjects above the 0.09 at baseline cutoff remained stable (Fig 8E). However, when we focused only on subjects below the cutoff at baseline (indicative of elevated Aβ deposition in the brain), and calculated the change in Aβ42/40 ratio from baseline to day 91, an increase was visible at the 35 and 85 mg CNP520 doses, while no change was observed in the placebo and low dose CNP520 groups (Fig 8F).

# Discussion

Therapies that target Aβ are the focus of efforts to develop a treatment for AD. Both BACE-1 inhibitors and anti-Aβ antibodies are the most advanced of these potential therapies in development. Up to now, however, both approaches have failed to show clinical benefit: A clinical trial of verubecestat in mild-to-moderate AD failed to show a benefit in cognition, despite close-to-maximal inhibition of Aβ production (Egan *et al*, 2018). In addition, the Phase III trials in prodromal/early AD of the BACE inhibitors verubecestat and lanabecestat have been stopped for futility at interim analysis, as announced by company press releases. One widely held hypothesis to explain this failure is that the intervention occurred too late during the course of the disease (Sperling *et al*, 2014). Patients at the stage of mild-to-moderate AD show substantial Aβ and tau (neurofibrillary) pathology as well as neuronal loss. Although less cognitively impaired, patients in the prodromal AD/mild cognitive impairment stage also show evidence of extensive Aβ pathology. While

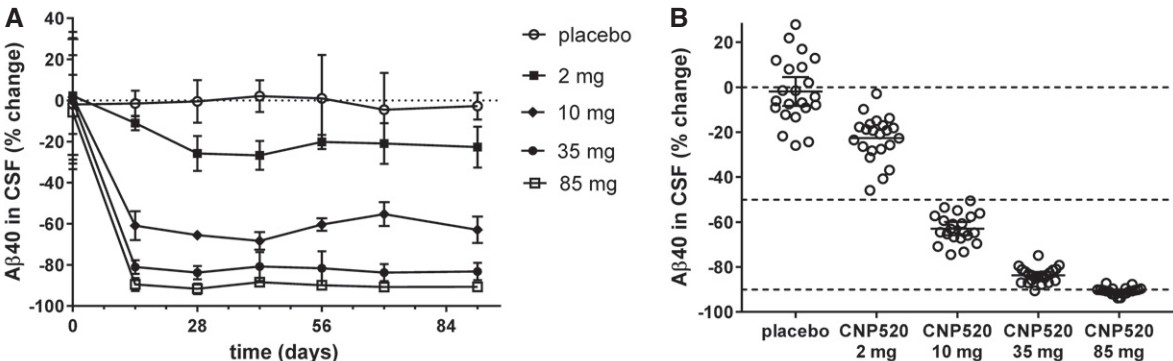

**Figure 7.  Effects of CNP520 administration for 3 months on Aβ40 in the CSF of healthy study participants ≥ 60 years of age.**

A    Changes of Aβ40 relative from baseline values up to 3 months of daily dosing. Shown are means ± SD, $n = 21–25$ for pre-dose and day 91, and $n = 2–7$ for the intermediate time points.

B    Dose–response for CSF Aβ40 at day 91 (24 h after last dose) and individual percent change from baseline, shown as mean and 95% confidence interval, $n = 21–25$ per treatment group.

treatment with verubecestat reduces levels of soluble Aβ forms, amyloid PET data indicate that deposited Aβ is reduced only slightly over 18 months, probably because of slow physiological clearance (Egan *et al*, 2018). This indicates that removal of soluble Aβ forms alone is not sufficient to halt the neurodegenerative processes, as long as substantial amounts of deposited Aβ remain present.

Evidence from genetic at-risk groups and otherwise cognitively unimpaired individuals suggests that progressive AD biomarker changes begin several years before cognitive impairment. One critical, detectable change is the accumulation of pathological Aβ species in the brain, beginning more than two decades before the onset of dementia. We predict that the benefits of a disease-modifying therapy targeting Aβ will be greatest during the preclinical stage of the disease, stopping Aβ accumulation in the brain before it has reached the plateau. At this stage, markers of neurodegeneration show little or no elevation, and cognition is unimpaired (Jack *et al*, 2013). However, prevention trials come with several challenges: (i), Identification of people at risk is difficult, since they have no or only minimal memory complaints; (ii), clinical parameters of cognition and memory change only very little over time at early disease stages, requiring very long trials with many participants; and (iii), not all individuals at risk will progress to clinical disease within their lifetime. It follows from these challenges that treatment of at-risk but asymptomatic individuals for a long time requires a drug with a particularly favorable safety and tolerability profile. Our data show that the BACE-1 inhibitor CNP520 has these characteristics.

CNP520 combines low nanomolar *in vitro* potency with functional selectivity over BACE-2, high brain penetration (with comparatively low peripheral exposure at active doses), and absence of any structural risks associated with the parent structure or its main metabolites.

BACE-2 is the closest evolutionary relative of BACE-1, with high structural similarity (Ostermann *et al*, 2006), and therefore a strong candidate for off-target effects (drug interaction with binding partners other than BACE-1). Although CNP520 displays only a threefold selectivity for BACE-1 over BACE-2 *in vitro*, long-term treatment did not result in hair depigmentation in mice or dogs, as described for BACE-1 inhibitors lacking BACE-2 selectivity (Cebers

*et al*, 2016; Kennedy *et al*, 2016; Shimshek *et al*, 2016). Skin concentrations of CNP520 have not been determined in dogs or humans. However, the estimated free plasma concentration of CNP520 (used here as a surrogate) is below or in the range of the *in vitro* IC$_{50}$ value for BACE-2 inhibition (Fig EV3). Regular skin examinations and centralized dermatological monitoring are implemented in the Generation Program as a precautionary action.

The importance of sparing the BACE-1-related enzyme CatD has become obvious from the reports on retinal degeneration in the rat after treatment with the BACE inhibitors LY2811376 and AMG-8718 (May *et al*, 2011; Fielden *et al*, 2015). Although both compounds showed CatD selectivity *in vitro* (> 3,000-fold for AMG-8718), their substantial accumulation in the acidic lysosome is most likely the reason for the impaired photoreceptor degradation and the associated pathology (Zuhl *et al*, 2016). It is likely that the excellent *in vitro* selectivity of CNP520 over CatD (> 20,000-fold) explains why accumulation of CNP520 in lysosomes does not lead to retinal pathology, as we have demonstrated in rodents and dogs.

Studies in BACE-1 KO mice that have identified a number of potentially adverse phenotypes, including myelin and muscle spindle alterations and retina atrophy, have raised concerns about the safety of BACE-1 inhibitors. In contrast, our long-term CNP520 toxicology studies in rodents and non-rodents did not show such changes. This strongly suggests that the phenotypes in BACE-1 KO mice are developmental in origin and are not caused by lack of BACE-1 in adult animals. Additionally, in contrast to a knockout situation, a BACE-1 inhibitor might not distribute equally between cellular compartments and is possibly more enriched in the acidic endosome. Therefore, it may inhibit BACE-1 in the endosome more than in the trans-Golgi network, where BACE-1 cleavage of physiological (non-APP) substrates occurs (Barao *et al*, 2015; Ben Halima *et al*, 2016).

We identified CNS effects and focal skeletal muscle atrophy in rats without functional effects as potential dose-limiting toxicities in our toxicological studies. To obtain an estimate of the CNP520 therapeutic window, we compared the systemic exposure at the clinical doses tested in the Phase IIa trial with the exposure in rats and dogs at the no observed adverse effect level (NOAEL). The safety margins at the highest dose tested in Phase IIa (85 mg) are as

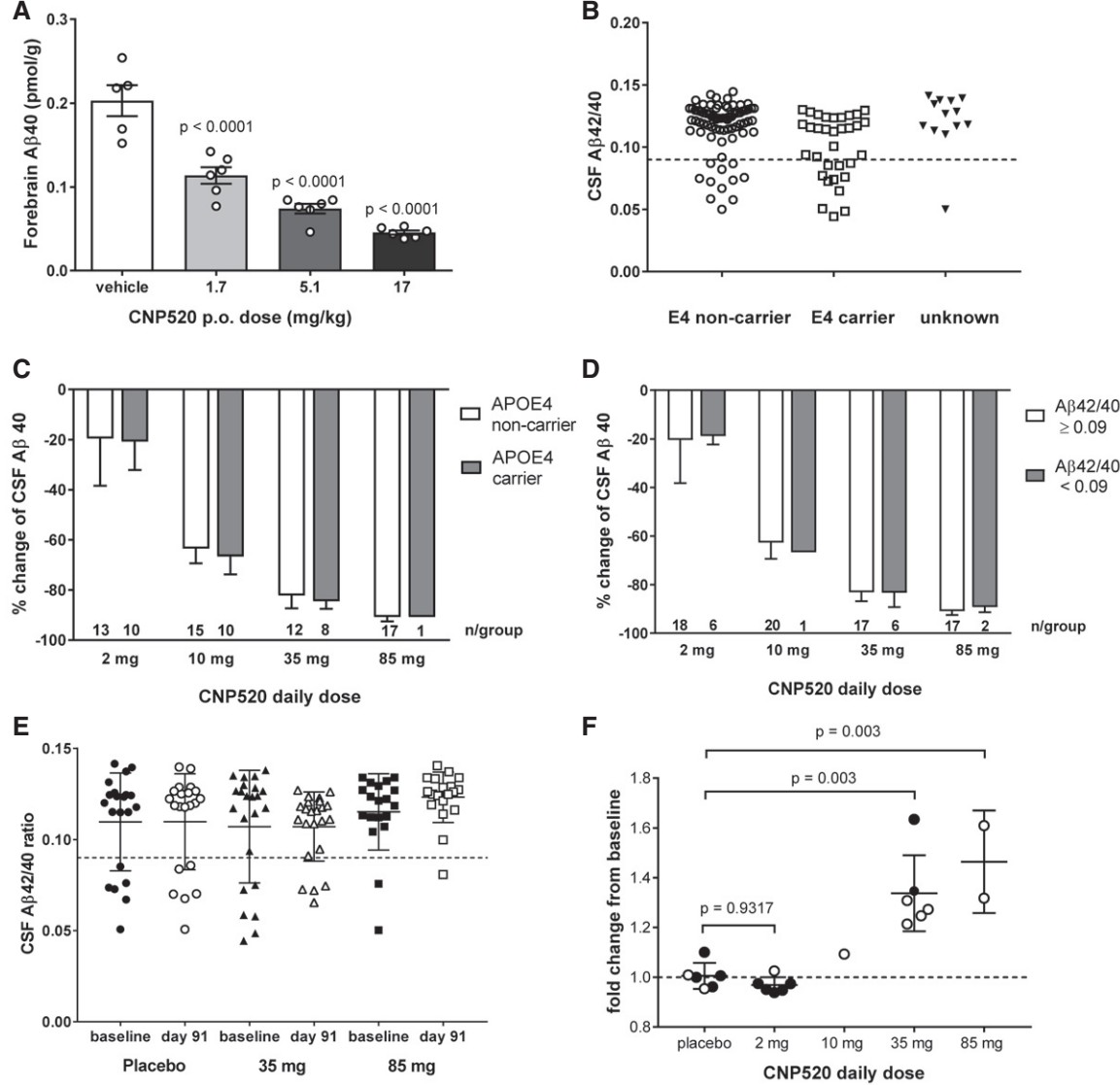

**Figure 8.   Effect of *APOE* genotype and baseline Aβ42/40 ratio on response to CNP520 treatment.**

A   Mouse forebrain Aβ40 levels in homozygous male and female *APOE4*-TR mice with and without CNP520 treatment, 4 h after dosing, shown is mean ± SEM, n = 6/group. Data were analyzed with one way ANOVA and Dunnett's multiple comparison test.

B   Distribution of *APOE4* genotypes in the Phase IIa study versus their baseline Aβ42/40 ratios. The dashed line at Aβ42/40 ratio of 0.09 represents a cutoff, which separates a normal ratio from an abnormally low ratio.

C   Comparison of CSF Aβ40 reduction in *APOE4* non-carriers and *APOE4* carriers (mean ± SD).

D   Comparison of CSFAβ40 reduction in participants with Aβ42/40 ≥ 0.09 and Aβ42/40 < 0.09 (mean ± SD).

E   Comparison of Aβ42/40 ratio at baseline and at day 91 for all participants in the placebo, 35 mg, and 85-mg dose group (mean ± SD, n = 25/group).

F   Change of Aβ42/40 ratio for all participants with Aβ42/40 ≤ 0.09 at baseline; bars represent mean ± SD. Full circles: *APOE4* carrier, open circles: *APOE4* non-carrier. Data were analyzed with one way ANOVA and Dunnett's multiple comparison test.

follows: ≥ 5-fold (male rats; clinical effects at ≥ 500 mg/kg/day), ≥ 7-fold (female rats; focal skeletal muscle atrophy without functional effects at 200 mg/kg/day), and ≥ 11-fold (female and male dogs; CNS effects at ≥ 30 mg/kg/day).

We found that CNP520 does not induce CMH, the equivalent of human ARIA events, in mice. No CMH events were observed for the BACE inhibitors NB-360 and verubecestat in APP-transgenic mice (Beckmann *et al*, 2016; Villarreal *et al*, 2017). Both these and our CNP520 results suggest that BACE-1 inhibition is not associated with brain vascular impairments.

The favorable CNP520 non-clinical safety and tolerability profile translates into early clinical studies. In our human studies described here, we observed no dose-limiting safety findings after multiple dose administration. All doses were safe and well tolerated; the observed adverse events were generally mild and transient. Ophthalmological, neurological, dermatological, cardiovascular, and cognitive monitoring have not raised safety flags.

Long-term safety human data have been generated for the BACE inhibitor verubecestat in the EPOCH study (NCT01739348) in patients with mild-to-moderate AD. Approximately 1,000 patients were

exposed to the drug for 18 months. Overall, the compound was well tolerated by most participants; however, neuropsychiatric symptoms and skin reactions occurred with higher frequency than for placebo (Egan *et al*, 2018). Although the selectivity, distribution, and metabolic profile of this compound is distinctly different from CNP520, potential class effects cannot be excluded, and the long-term safety and tolerability of CNP520 in humans needs to be demonstrated.

The primary pharmacodynamic effect (reduction of soluble Aβ40 and Aβ42 in brain and/or CSF) was observed in mice, rats, and dogs at similar CNP520 exposure, and this translated to humans very closely: The free $ED_{50}$ in dog blood was the same as the free $ED_{50}$ in human plasma (3.3 nM). In human studies, CNP520 robustly reduced human CSF Aβ in a dose-dependent manner, confirming the potency observed in non-clinical *in vitro* and *in vivo* studies. Furthermore, long-term treatment with CNP520 translated to substantially reduced plaque load and plaque-associated neuroinflammation (as measured by the number of activated microglia and astrocytes) in the brains of APP23 mice. In addition to toxic forms of Aβ, the Aβ-associated neuroinflammatory processes are one of the key drivers of neuronal dysfunction and cognitive deficits in AD (Heneka & O'Banion, 2007). Studies with NB-360, another oxazine BACE-1/2 inhibitor, in transgenic mice have shown beneficial effects on AD-relevant downstream markers tau and neurofilament light chain, as well as neuronal behavior restoring impaired brain connectivity (Bacioglu *et al*, 2016; Keskin *et al*, 2017; Schelle *et al*, 2017). We are aware of the limitations of studies in APP-transgenic mice for translation to the human disease. Whereas such animal data are useful to demonstrate the general direction of the treatment effect, they may not be able to predict the size of the treatment effect in the human AD brain, given the substantial differences between the slowly developing human disease and the highly accelerated Aβ pathology in the rodent models. In particular, in our mouse studies, we have not observed a reduction of Aβ load below the baseline, which would indicate a net removal of Aβ from the brain. We assume that plaque dissolution is a slow process that cannot be observed within the study timelines of a few months. Therefore, our mouse studies primarily provide support for the prevention approach. While a modest reduction on Aβ PET signal was observed with verubecestat, it remains to be seen whether a multi-year treatment with a BACE-1 inhibitor will result in a relevant reduction of plaques in humans.

The Generation Program, which is evaluating the efficacy of CNP520 in preclinical AD, enrolls people at risk of developing symptoms of AD based on the presence of the *APOE4* allele and, in *APOE4* heterozygotes, the additional presence of fibrillary Aβ. Our Phase IIa study population included participants who are *APOE4* carriers and/or have already elevated brain Aβ, which allowed us to compare the CNP520 pharmacodynamic effect across subgroups. As an indicator of increased Aβ deposition in the brain, we used reduction in the CSF Aβ42/40 ratio below normal, which corresponds robustly to $^{18}$F-flutemetamol amyloid PET imaging of cortical Aβ, the gold standard for determining Aβ levels in the brain (Janelidze *et al*, 2016; Pannee *et al*, 2016). When analyzing the baseline CSF Aβ42/40 ratio data, we found a larger group with a normal Aβ42/40 ratio that is above 0.09 and a smaller group below the cutoff of 0.09, indicating elevated brain Aβ. We found participants with a biomarker signature for ongoing Aβ deposition both in the *APOE4* non-carrier and in the carrier group, but the proportion was higher in the carrier group, in line with reports documenting an increased

risk of early Aβ deposition in *APOE4* carriers (Jansen *et al*, 2015; Hollands *et al*, 2017). The response of participants with normal CSF Aβ42/40 ratios and of those with the lower ratio to CNP520 treatment did not differ across dose groups. Furthermore, a trend toward normalization of the CSF Aβ42/40 ratio was observed in subjects with a low ratio at baseline after treatment with CNP520 for 13 weeks, with doses leading to approximate 80% CSF Aβ lowering. This is a first hint that treatment with CNP520 reduces Aβ deposition in the brain in subjects with early Aβ pathology. While the data suggest that we have chosen a relevant dose range for long-term studies, we need additional data, including amyloid PET measurements from a larger patient cohort, to demonstrate an effect of CNP520 treatment on brain fibrillary Aβ load.

Inhibition of BACE-1 is a promising disease-modifying strategy for AD, validated by human genetics (Jonsson *et al*, 2012). The concept of BACE-1 inhibition for the treatment of AD has gained significant momentum since the identification of low-molecular-weight, highly potent and brain-penetrating compounds derived from a cyclic amidine core structure (Zhu *et al*, 2010), such as CNP520. Currently available data suggest that this class of compounds is in general well tolerated and can be given to humans over extended periods. Although experience from long-term dosing in large populations is not yet available, CNP520, with its lack of aniline metabolite formation, its functional selectivity over BACE-2, and good safety window, appears suited to long-term treatment. Based on our understanding of the development of AD pathology over time, we expect that intervention during the phase of Aβ buildup might have a higher probability of success than treatment started in the established dementia stage. The Generation Program will provide information on the therapeutic value of CNP520 in the prevention of AD.

## Materials and Methods

### Materials

CNP520 was synthesized at Global Discovery Chemistry and at Technical Research and Development departments at Novartis, Basel, Switzerland, and Shanghai, People's Republic of China, according to Badiger *et al* (2013). A capsule form based on wet granulation of standard pharmacopoeial excipients and the micronized drug substance was selected as the first clinical form. Good process performance was observed during clinical manufacturing at batch size up to 40,000 units. The shelf life of the form was 24 months when kept refrigerated.

### *In vitro* assays

Enzyme inhibition assays to determine the potency of CNP520 on various aspartyl proteases were done as described using recombinant catalytic domains and fluorescence-quenched peptide substrates (Neumann *et al*, 2015). Cellular Aβ release assays were done in CHO cells stably transfected with human wild-type and human APP were done as described (Neumann *et al*, 2015). Determinations of logD, pKa, and binding to brain homogenates were done internally at Novartis, Basel. Plasma protein binding was determined by equilibrium dialysis using diluted plasma from the corresponding species. MDR-1-MDCK cells were obtained from Prof.

A. Berns, Netherlands Cancer Institute, Amsterdam NL, and bi-directional transport measurements were done as described (Rueeger et al, 2011). Receptor binding assays were done against an internal receptor panel and externally at Ricerca Biosciences, Peitou, Taiwan. In vitro binding and cellular patch clamp assays for hERG were done at Preclinical Safety, Novartis, Basel.

### Animals

Experiments were carried out in accordance with the guidelines of the Swiss Federal and Cantonal veterinary offices for care and use of laboratory animals or the Canadian Council on Animal Care (Ottawa, Canada), and the ARRIVE guidelines. Studies described in this report were approved by the Swiss Cantonal Veterinary Authority of Basel City, Switzerland, and performed according to animal license numbers BS-2063, BS-2077, and BS-1094. Methods employed in PK/PD studies in rats and dogs are described (Neumann et al, 2015).

Male rats (Sprague Dawley, 3–4 months old) and C57BL/6J mice (male, 5 months of age) were sourced from Charles River Laboratories (France). Animals were maintained group-housed under standard conditions in temperature and humidity controlled rooms under a 12/12 light/dark schedule. Standard laboratory rodent food and tap water were available ad libitum. The dog PK/PD study was conducted in male Beagle dogs (4 months of age) at Charles River Laboratories (Senneville, Canada). Dogs were housed individually under climate-controlled conditions in caging that provided room for exercise. Lighting was provided for 12 h/day. Animals were fed ad libitum with a certified commercial canine diet for 6 h daily, and water was available ad libitum.

Female and male APP23-transgenic mice (B6.D2-Tg (Thy1App) 23/1Sdz, available from Jackson Laboratory Stock No: 030504) were used. APP23 mice express human APP751 with Swedish mutations, under the control of the murine Thy-1 promotor (Sturchler-Pierrat & Staufenbiel, 2000). Male and female transgenic homozygous APOE4-TR (B6.129P2-Apoetm3 (APOE*4)Mae N8, Taconic, Model 001549, 3–5 months old) were ordered from Taconic. BACE-1 (B6,129T2-TgH(Bace1)1Goe) and BACE-2 (B6,129P2-TgH(Bace2)1Leu) KO mice were received from the laboratory of B. De Strooper (Dominguez et al, 2005). BACE-1 and BACE-2 KOs were back-crossed at least six generations into C57BL/6 (> 98%).

Mice were single-housed and maintained under standard conditions in temperature and humidity controlled rooms under a 12/12 light/dark schedule, with lights on at 05:00 AM. Cage bedding consisted of sawdust and a red Perspex house (Nalgene, Thermo Fisher Scientific, Waltham, MA, USA). Nesting materials (Nestlet, Ancare, Bellmore, NY, USA) and a wooden gnawing-block were supplied in each cage. Tap water and standard laboratory rodent food (± compound) were available ad libitum.

### Treatment

Compounds were either mixed in feed or given as a suspension orally. CNP520 and NB-360 were formulated as a suspension. Vehicle or compounds were given orally in a volume of 10 ml/kg once daily (mornings) for 8 weeks. Vehicle: 0.1% Tween-80 in 0.5% methylcellulose in water. CNP520 was dosed at 0.3 g/kg (high dose) and 0.03 g/kg (low dose) pellet (Provimi Kliba, Kaiseraugst, Switzerland). The vehicle group was treated with food pellets containing no compound. The β1 antibody was dosed 0.5 mg/kg body weight, once weekly intraperitoneally, diluted in 90 nM NaCl, 50 nM Tris pH 7.1.

### Quantification of CNP520 and metabolites

In the clinical studies, CNP520 concentrations were determined in plasma, urine, and CSF with a validated LC-MS/MS method. The lower limit of quantification (LLOQ) was 1 ng/ml in plasma and 2 ng/ml in urine and CSF, except for the 3-month study where the method for CSF was more sensitive (LLOQ 0.1 ng/ml). Plasma PK parameters were calculated using actual recorded sampling times and non-compartmental methods with Phoenix WinNonlin version 6.4 (Certara, Inc., Princeton, NJ, USA).

### Quantification of APP metabolites

Electrochemiluminescence immunoassays (Mesoscale Diagnostics, Rockville, MD, USA) were used to determine Aβ38, Aβ40, Aβ42, sAPPα, and sAPPβ. For studies in normal mice and rats, the endogenous Aβ40 was extracted from the brain homogenate with 1% Triton X-100, and the 4G8-based assay kit was used for quantification according to the manufacturer's instructions. CSF was collected from the cisterna magna and also analyzed using the 4G8 kit. For studies in dogs, CSF was collected via a ventricular port, and the 6E10-based Mesoscale Diagnostics kit for Aβ was used. For studies in APP23 mice, Aβ40 and Aβ42 were determined after extraction with 70% formic acid and subsequent neutralization.

Human CSF was collected from an indwelling spinal catheter inserted into the lower spinal canal (in the single ascending dose (SAD) PD first in human (FIH) study) or by a lumbar puncture (in the multiple ascending dose (MAD) PD Phase I study and Phase IIa study), and by trained personnel at the site and as per their standard operating procedure (SOP). CSF Aβ38, Aβ40, Aβ42, sAPPα, and sAPPβ were quantified in human CSF with 0.2% (v/v) Tween®-20 using validated sandwich-based multiplexed electrochemiluminescence methods (MesoScale Discovery, Rockland, MD, USA).

### Toxicology studies

In vivo animal toxicity studies were performed according to ICH guidance for industry M3(R2) and conformed to the Institutional Animal Care and Use Guidelines for the contract laboratory and sponsor. Wild-type (WT) mice (Jic:CB6F1-nonTgrasH2@Jcl (wild type)) from Taconic Farms, 11 weeks of age, were given 0 (vehicle), 30, 60, or 120/100 mg/kg/day CNP520 by daily oral gavage for 4 weeks. Rats (HsdHan™:WIST) from Harlan UK, 9 weeks of age, were given 0 (vehicle), 5, 30, or 200 mg/kg/day CNP520 by daily gavage for 26 weeks. Recovery was assessed after a 12-week treatment-free period. Beagle dogs from Marshall BKU UK, 11 months of age, were given 0 (vehicle), 5, 15, or 30 mg/kg/day for 39 weeks. Vehicle consisted of 0.5% (w/v) methylcellulose (1,500 cPS) in 0.1% (v/v) polysorbate 80 (Tween®-80) in reverse osmosis water. At necropsy, tissues were immersion fixed in 10% neutral-buffered formalin or modified Davidson's solution and then processed by routine methods to paraffin block and hematoxylin–eosin (H&E)-stained histologic slides.

KO mice and their heterozygote and WT controls were evaluated at 5 months of age for genotype-related changes in routine clinical chemistry, hematology, and light microscopic evaluation of a full

set of tissues. In addition, retina was assessed by transmission electron microscopy, as described in Shimshek *et al* (2016), and quantitative assessment by g-ratio calculation using automated image analysis of peripheral nerves.

### APOE4 genotyping

Whole blood was collected into a 6-ml plastic K2-EDTA spray-dried vacutainer (BD Cat. # 367863). DNA from the FIH and Phase IIa study samples was extracted using AutoPure LS 98 Automated Nucleic Acid Purification Instrument (Qiagene) and Chemagen magnetic beads, respectively. DNA samples were stored at −70°C before analysis. Genotype analysis was performed by TaqMan and bi-directional Sanger sequencing. The *APOE4* alleles were determined by rs429358 and rs7412.

### Clinical study design and participants

The studies described in this manuscript were conducted to characterize the safety, tolerability, and PK and PD effects of CNP520 in healthy human subjects. All completed clinical studies were randomized, double-blind, and placebo-controlled, except two open-label studies to assess the relative bioavailability of different CNP520 formulations and potential drug–drug interactions. CNP520 is currently being tested in Generation studies I and II, which are registered with Clinical Trials.gov under the identifiers NCT03131453 and NCT02565511. For further details, see the Appendix Table S10 in Appendix and the accompanying clinical protocols synopsis (Appendix Table S11).

### Clinical study approval

All the studies were conducted in accordance with ICH Good Clinical Practice guidelines, the Department of Health and Human Services Belmont Report, and the Declaration of Helsinki. They were approved by the NRES Committee South Central Berkshire, Bristol, UK, the State Office of Health and Social Affairs Berlin, Germany, The commissie voor Medische Ethiek Institutional Review Board Antwerp, Belgium, the Medisch Ethische Toetsings Commisie, Assen, the Nederlands, and Quorum Review IRB Seattle, WA, USA (EUDRACT number: 2013-005576-18). Written informed consent was obtained from each participant before any study procedures.

### Statistics

Animals were pseudo-randomly allocated to treatment groups. With the exception of the hair discoloration study, there was no blinding in the acute and chronic animal studies. In the hair discoloration study, the observer for hair color changes was blinded and did not know the allocation of the mice into treatment or control groups. The histological plaque load and associated neuroinflammation were statistically evaluated by one-way ANOVA Dunnett's multiple comparisons test. Each data set was tested for normality of distribution, and appropriate parametric or non-parametric) tests were used. Reported *P* values were adjustment for multiplicity of testing. Only groups with similar variance were compared. Clinical studies were double-blind, randomized studies. Randomization numbers were

### The paper explained

#### Problem

Alzheimer's disease (AD) is the most prevalent neurological disorder, with only symptomatic treatments available. Drug discovery efforts have focused on amyloid-β deposition in the brain, one of the key pathological features of AD. However, multiple clinical trials that targeted Aβ have failed when conducted in patients at early or moderate dementia stages.

#### Results

Aβ-targeting treatments need to start in the asymptomatic disease state, to prevent Aβ accumulation to toxic levels. Potential medications for prevention treatment need to have an exceptional safety profile, allowing their long-term use in pre-dementia patients. The BACE-1 inhibitor CNP520 meets these requirements, due to its superior selectivity profile, its high brain penetration, and absence of potentially toxic metabolites. The pharmacodynamic activity and safety of CNP520 were demonstrated in acute and chronic animal studies, as well as in Phase I and Phase IIa clinical studies.

#### Impact

The data presented here allowed the start of two long-term clinical prevention studies in subjects at risk for Alzheimer's disease (Generation studies I and II, currently recruiting). These trials will provide information whether or not prevention treatment can delay or stop the disease progression and the onset of dementia due to AD.

generated to ensure that treatment assignment was unbiased and concealed from subjects and investigator staff. A randomization list was produced by Novartis Drug Supply Management using a validated system that automated the random assignment of treatment arms to randomization numbers in the specified ratio. The randomization scheme for subjects was reviewed and approved by a member of the Novartis IIS Randomization Group. For clinical Phase I and II studies, all subjects having received at least one dose of study drug were included in the analysis of safety data. All subjects having received at least one dose of study drug and with no major protocol deviations with relevant impact on the analysis of pharmacodynamic (PD) evaluation were included in the analysis of PD data (Aβ, sAPP). All biochemical analysis of clinical samples was done blinded. Unblinding was done when analysis was fully completed and only by the study statistician.

Descriptive statistics provided for safety and PD data included *n*, mean, SD or SEM, median, and range. For Aβ, sAPP data, percent changes from baseline were calculated at each post-baseline time point and were analyzed using an ANCOVA model with treatment (CNP520 dose or placebo) as factor and baseline value as covariate. No adjustment of alpha level for multiplicity was done as the inferential analyses were considered exploratory.

## Data availability

The data for the CNP520-BACE-1 X-ray structure are available at the Protein Data Bank, https://www.rcsb.org/ under the accession code 6EQM.

**Expanded View** for this article is available online.

## Acknowledgements

We thank C. Rolando, U. Stauss, J. Zadrobilek, S. Kläusler, K. Barker, S. Jacquier, M. Hediger, J. Kohlstedt, P. Köninger, T. Zimmermann, I. Brzak, V. Trappe, T. Dittmar, S. Zurbruegg, P. Wipfli, R. Endres, L. Hoffmann, S. Desrayaud, T. Delemonte, G. Laue, and M. Jivkov for skillful assistance during compound preparation and testing. We thank R. Hemmig for the provision of refolded BACE-1 and E. Wirth for BACE-1 crystal preparation. The help of the Swiss-Light Source, Paul Scherrer Institute, Villingen, Switzerland, is acknowledged. Y. He and Y. Pluess-Li conducted APOE4 genotyping. We thank D. Feuerbach and R. Dolmetsch for critically reading the manuscript and helpful comments. The authors thank Joan Smyth (Ph.D.) of Novartis Ireland Limited, Dublin, Ireland, for providing editorial support, which was funded by Novartis Pharma AG, Basel, Switzerland, in accordance with Good Publication Practice (GPP3) guidelines (https://www.ismpp.org/gpp3).

## Author contributions

UN, MS, RML, RM, SJV, HR, KH, and MT-B contributed to the concept development, drug design, *in vitro* testing, and optimization. RML, RM, SJV, KH, and HR performed medical chemistry work. MU, M-LR-D, AG, and CLL designed and conducted clinical studies. LHJ, DRS, LP, WF, VD, and BV performed preclinical pharmacology studies. KB and HS performed preclinical DMPK studies. GH performed clinical PK studies. CK and AH performed preclinical safety studies. AA and SK performed clinical biomarker and genotyping studies. NP, SV, and OJD performed data analysis of clinical studies. NB performed preclinical MRI studies. J-MR did X-ray crystallography studies. BG and RR performed formulation studies for clinical material. UN and CLL wrote the manuscript with input from all authors.

## Conflict of interest

During the time this work was done, all authors have been employees and shareholders of Novartis Pharma AG, Basel, Switzerland. U. Neumann, Rainer M. Lueoend, Rainer Machauer, Siem J. Veenstra, Konstanze Hurth, Heinrich Rueeger, Marina Tintelnot-Blomley, and Cristina Lopez Lopez are inventors in granted or pending patent applications.

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
