## [Review Process File · EMBO Molecular Medicine]

The BACE-1 inhibitor CNP520 for prevention trials in Alzheimer's disease

Ulf Neumann, Mike Ufer, Laura H. Jacobson¹, Marie-Laure Rouzade-Dominguez, Gunilla Huledal, Carine Kolly, Rainer M. Lüönd, Rainer Machauer, Siem J. Veenstra, Konstanze Hurth, Heinrich Rueeger, Marina Tintelnot-Blomley, Matthias Staufenbiel, Derya R. Shimshek, Ludovic Perrot, Wilfried Friauff, Valerie Dubost, Hilmar Schiller, Barbara Vogg, Karen Beltz, Alexandre Avrameas, Sandrine Kretz, Nicole Pezous, Jean-Michel Rondeau, Nicolau Beckmann, Andreas Hartmann, Stefan Vormfelde, Olivier J. David, Bruno Galli, Rita Ramos, Ana Graf, Cristina Lopez Lopez

Review timeline:	Submission date:	09 May 2018
	Editorial Decision:	21 June 2018
	Revision received:	20 July 2018
	Editorial Decision:	21 August 2018
	Revision received:	23 August 2018
	Accepted:	24 August 2018

Editor: Céline Carret

Transaction Report:

1st Editorial Decision

21 June

Thank you for the submission of your manuscript to EMBO Molecular Medicine. We have now heard back from the three referees whom we asked to evaluate your manuscript.

You will see from the comments pasted below that all three referees are supportive of publication and only have minor (and overlapping) comments that should nevertheless be addressed in the revised version of the paper.

We would welcome the submission of a revised version within three months for further consideration and would like to encourage you to address all the criticisms raised as suggested. Please note that EMBO Molecular Medicine strongly supports a single round of revision and that, as acceptance or rejection of the manuscript may depend on another round of review, your responses should be as complete as possible.

***** Reviewer's comments *****

Referee #1 (Comments on Novelty/Model System for Author):

Although CNP520 appears to be effective at lowering Abeta and safe in animal models and early phase human clinical trials, it would be helpful to understand mechanisms and toxicity of BACE-1 depletion in post-development, adult mice, such as an inducible conditional BACE-1 KO mouse line. It is unclear if these animals exist. This is certainly not a necessity for this publication but just a suggestion for the future, if available.

Referee #1 (Remarks for Author):

This paper contains many experimental results in vitro, in vivo in animal models and in vivo in human clinical trials. Therefore, it may be too large for a Short Report.

Neumann and colleagues (from Novartis) report the structure, selectivity, PD/PK, distribution, efficacy and safety of their BACE-1 inhibitor, CNP520, in preclinical models (in vitro and in vivo) and safety and target engagement in humans. CNP520 is being developed by Novartis for the prevention of AD, as they believe that lowering A β production at the earliest stage of AD pathogenesis, prior to downstream (subsequent) changes in neuroinflammation and neurodegeneration, is likely to be most effective. The authors state the need for good brain penetration, strong selectivity of BACE-1 over BACE-2, and avoidance of any metabolites that might result in mutagenic or toxigenic events. Here, the authors report CNP520's structure, and show that brain delivery is achieved without much efflux. CNP520 is 3-fold more selective for BACE-1 than BACE-2 and 20-fold more selective for BACE-1 than cathepsins. The drug appears to be safe and non-addictive and did not cause changes in hair color in beagles or rats. Metabolism studies indicated the aniline metabolites, associated with genotoxicity, were not generated. No changes were observed in retina, and CNP520 did not induce cerebral microhemorrhages (evaluated by MRI imaging) in aged APP23 mice with CAA. Robust and long-lasting lowering of A β protein was achieved in beagles, rats, APP23 mice and humans (CSF). There was no difference in the response of APOE4 carriers to CNP520 compared to other APOE genotype carriers.

Overall, this is a very nicely constructed paper containing a tremendous amount of well-organized, strong data to support the move forward to clinical trials using CNP520 for the prevention of AD, as is now underway in the GENERATION trial.

Comments:

1. P10: "CNP520 is highly BACE-1-specific, with the exception of BACE-2". Elsewhere, the data show that CNP520 is 3-fold more selective for BACE-1 than BACE-2. Is this enough? The functional and safety studies do not seem to indicate major problems. What, in particular, will be done to monitor for BACE-2 inhibition in the human clinical trials?
2. P8: Non-specific binding of CNP520 to plasma proteins - which one(s)?
3. Eighteen month-old APP23 mice, bearing CAA, were treated daily with CNP52- (55 mg/kg) for 3 months. MRI showed no increase in cerebral microhemorrhage (CMH). Were the brains examined pathologically for CMH? If so, what were the results? If not, why? MRI is not the most sensitive measure of CMH, especially in a terminal study. Iron staining (e.g., hemosiderin) would allow localization of CMH in specific brain regions and would help determine if CMH was elevated in blood vessels containing CAA.
4. Was CAA reduced by CNP520 treatment in aged APP23 mice?
5. CNP520 was tested in beagle dogs. How old were these animals? Did they have plaque deposition at the start of the study? Did they have any CAA? (Page 13: how many doses?) Was this a terminal study? If not, would these dogs be expected to have plaque deposition at the age tested (for PK/PD)?
6. Is beagle BACE-1 homologous to human BACE-1?
7. P9: Dose-limiting non-specific CNS effects (including impaired mobility and tremor) occurred in dogs at > 30 mg/kg/day and in rats at > 500 mg/kg/day. The dose limit in dogs seems relatively low. Did the dogs recover? Is this an acceptable risk? Have any humans shown such side effects in the clinical trials to date?

8. Do APP23 mice have CAA at 12-14 mo of age (the start of treatment in one study)? Did CNP520 reduce (or increase) CAA in this study? Was CNP520 treatment associated with any increase in CMH?
9. It is interesting that there was a differential effect of CNP520 on microglia and astrocytes in the APP23 mice orally treated (in food) from ~13 mo to ~19 mo of age. Insoluble A β , plaques and sAPP β levels were reduced while sAPP α was increased, confirming a lowering of amyloidogenic processing of APP. There was a reduction of plaque-associated microglia (Iba-1, which is a microglia and macrophage marker), but no change in microglia distant from plaques. However, astrocytes both near and away from plaques were reduced. Does CNP520 have any direct effects on astrocytes (or microglia)? A further discussion of this interesting difference is warranted.
10. P16: One subject on 86 mg CNP520 dropped out of the study due to "an AE of global amnesia (moderate), possibly due to transient ischemic attacks, according to a neurologist's examination". Was this AD deemed drug-related? Did this person, or any other subject experience any problems with mobility or other CNS effects?
11. In Figure 8F, it appears possible that in plaque-bearing individuals (with a low CSF A β 42/40 ratio), there might be a possible effect of APOE4 genotype. Why were only 2 APOE4 carriers included in the 3 highest dose groups?
12. P22-23: Many of the phenotypes in BACE-1 KO mice are considered developmental. Do BACE-1 inducible conditional KO mice exist? If so, is BACE-1 KO post-development associated with any of these deleterious phenotypes (e.g. myelin and retinal changes)? [These animals may or may not exist but would be useful here.]
13. The authors should be sure to update their discussion regarding BACE-1 inhibition based upon any recent clinical trial announcements.
14. P28: "Methods employed in PK/PD studies in dogs and rats are described in Neumann (Neumann et al, 2015b)." Remove the first "Neumann"?
15. Lastly, have the authors detected any gender-specific effects of CNP520 in any of their 5 human clinical trials thus far? Is there any reason to believe that the drug's efficacy or side effects might be different in men and women?

Referee #2 (Comments on Novelty/Model System for Author):

Rats mice dogs humans used in this study are relevant. No ethical concerns.

Referee #2 (Remarks for Author):

Here, the authors report for the first time the structural, pharmacokinetic, pharmacodynamic, and toxicity data for CNP520, a novel BACE1 inhibitor for Alzheimer's disease (AD). In this very comprehensive study, they show that CNP520 has favorable selectivity and safety characteristics that make this drug particularly attractive for the prevention of AD prevention, which would require an exquisitely safe therapeutic agent. One of the most compelling and unique characteristics of CNP520 is that it is ~3-fold more selective for BACE1 over BACE2, and that its concentrations in the skin and other tissues are low compared to the BACE2 IC₅₀. Together, these features of CNP520 result in absence of the hypopigmentation associated with other BACE1 inhibitors now in clinical trials, which are equipotent at inhibiting BACE1 and BACE2. Other desirable features of CNP520 are that the drug has equivalent effects in ApoE4 carriers and non-carriers, and that it does not cause micro-hemorrhages like anti-A β antibody treatment. CNP520 treatment in APP transgenic mice resulted in dramatic reduction of amyloid pathology and gliosis, and in humans increased the A β 42:A β 40 ratio in individuals with amyloid accumulation, suggesting that the drug had slowed amyloid deposition. CNP520 also does not inhibit cathepsin D like some other BACE1 inhibitors, and in line with this observation the drug did not cause retinal pathology or other cathepsin D-related side effects.

As mentioned above, CNP520 is unique among the current BACE1 inhibitors, particularly for its selectivity for BACE1 over BACE2 and safety profile, making it a compelling candidate drug for long-term use for the prevention of AD in presymptomatic individuals. In fact, CNP520 is currently in a unique AD prevention trial in ApoE4 carriers (the Generation study). Thus, CNP520 is among the first BACE1 inhibitors to test a new prevention paradigm for AD. The authors should be congratulated on a very comprehensive and informative study that makes a significant contribution to the AD therapeutic field. The results are rigorous and robust, and the manuscript well written.

I have only a few minor questions, the answers to which I suggest the authors may consider, because they may be informative for the reader and add depth and more nuance to the Discussion. Foremost, it would be highly informative to put CNP520 into a more extensive context with the other BACE1 inhibitors in clinical trials. Some questions that come to mind are the following. How do the structure and drug properties of CNP520 compare to those of the other major inhibitors? Perhaps a comparative table or diagram would be useful. From what has been published for the major inhibitors, how do their adverse events (AEs) compare to those of CNP520? For CNP520 and the other inhibitors, are the AEs likely to be on-target or off-target? If off-target, could the differences in structures between the compounds be responsible for the off-target AEs? Comparison of CNP520 to verubecestat may be most informative in regard to these questions, as more data has been published on the latter than the other inhibitors. In particular, the authors should comment on these questions in relation to the recently published Egan et al (2018) New England J. of Medicine 378;18:1691 article on the EPOCH verubecestat trial results. Also, it would be informative for the authors to comment on the recently announced failure of JNJ-54861911 due to liver toxicity.

Referee #3 (Remarks for Author):

In this manuscript, the authors reported the pharmacological characterization of BACE1 inhibitor CNP520 and the results of clinical trials. CNP520 effectively reduced the CNS Abeta levels in animal models including APP transgenic mouse. CNP520 treatment also decreased the Abeta reduction in the CSF of healthy elderly without apparent adverse effect. These data support that CNP520 is a candidate drug for prevention trials. An impressive array of methodologies and models are utilized in this study and seemed to be executed properly. I would recommend authors to include following points and discuss appropriately in the discussion section.

1. Recent results of clinical trials of BACE1 inhibitors

Authors should include the reports regarding the trial of verubecestat (Egan et al., NEJM 2018) and lanabecestat (Alzforum or related website), and discuss about efficacy and the effect on model animals (reduction in CSF Abeta and Abeta deposition). Especially, result of verubecestat suggested that, in contrast to amyloid plaques in the brains of rodent AD model (as shown in this manuscript and the other compounds), decreased production of monomer Abeta did not lead to effective remodeling/clearance of senile amyloid developed in human brain. Please provide possible different characters at molecular level between rodents and humans and discuss this issue in appropriate manner.

2. Effect of CNP520 on hair pigmentation

Results clearly suggested that the skin concentration of the compound significantly contributed the complete absence of hair depigmentation. If available, authors should show the data of PMEL17 processing in the skin of CNP520-treated mouse. Also, please provide the concentration of CNP520 in other model (i.e., dogs) as well as humans to strengthen the idea that distribution of CNP520 is a crucial factor.

1st Revision - authors' response

20 July

Reviewer 1

1. P10: "CNP520 is highly BACE-1-specific, with the exception of BACE-2". Elsewhere, the data show that CNP520 is 3-fold more selective for BACE-1 than BACE-2. Is this enough? The functional and safety studies do not seem to indicate major problems. What, in particular, will be done to monitor for BACE-2 inhibition in the human clinical trials?

Answer : Our data suggest that the lack of BACE-2 specific side effects results from the combination of higher IC50 and an altered tissue distribution as described on p 11/12. This conclusion is based on a mouse study only, but is supported by the absence of visible hypopigmentation in dogs. In the Discussion section on p 23, a sentence has been inserted describing the regular skin examinations implemented in the Generation Program.

2. P8: Non-specific binding of CNP520 to plasma proteins - which one(s)?

Answer: The term “plasma protein” here refers to the native mix of proteins (albumins, globulins, glycoproteins, lipoproteins) that is found in the plasma of the corresponding species. Plasma preparations from the corresponding species are being used to determine total plasma protein binding, without making efforts to differentiate the binding to the individual proteins. A new sentence was inserted in Methods (p 29) to describe this better.

3. Eighteen month-old APP23 mice, bearing CAA, were treated daily with CNP52- (55 mg/kg) for 3 months. MRI showed no increase in cerebral microhemorrhage (CMH). Were the brains examined pathologically for CMH? If so, what were the results? If not, why? MRI is not the most sensitive measure of CMH, especially in a terminal study. Iron staining (e.g., hemosiderin) would allow localization of CMH in specific brain regions and would help determine if CMH was elevated in blood vessels containing CAA.

Answer: Indeed, mouse brains have been examined histo-pathologically to investigate effects of CNP520 treatment on cerebral microhemorrhages. The text on page 12 has been updated, and results of the histological examination are now shown in EV Figure 1B in a tabular format including a table legend. The histological data support the notion that CNP520 treatment does not increase the frequency of CMH.

4. Was CAA reduced by CNP520 treatment in aged APP23 mice?

Answer: Investigation on the effect of CNP520 treatment on CAA in APP23 mice has been recently performed, and first data indicate reduction of vascular A β similar to the effect on plaques. Given the already very long current manuscript, we are afraid that there is not enough space available to describe and discuss the data in appropriate detail, and plan to publish these results separately after more extensive analysis.

5. CNP520 was tested in beagle dogs. How old were these animals? Did they have plaque deposition at the start of the study? Did they have any CAA? (Page 13: how many doses?) Was this a terminal study? If not, would these dogs be expected to have plaque deposition at the age tested (for PK/PD)?

Answer: This was a standard PK/PD study and the dogs were not considered a disease model. The intention was solely to measure the CNP520 exposure and the effects on soluble CSF and plasma A β after a single oral dose of CNP520 to establish a PK/PD relationship to predict the dose for toxicology studies and the human active doses. The beagle dogs used in the study were young animals (3 months of age). At this age neither plaque deposition nor CAA is expected to be present in these animals, but this was not investigated. It is known from the literature that old dogs brains (older than 10 years) often show amyloid- β deposition in parenchyma and vasculature (Schmidt, F, et al, J. Neuropathol. Exp. Neurol. 2015 Sep; 74(9): 912-23, Nesić et al, Vet Q 2017, Dec, 37, 1-7). The study was not terminal, brains or other organs were not investigated. Wording on p 13 has been updated to reflect age of the dogs and the single dosing regimen.

6. Is beagle BACE-1 homologous to human BACE-1?

Answer: Canine BACE-1 has 82.4% sequence identity to human BACE-1 (Swissprot database, entries P56917 for human BACE-1 and F1P9Q0 for canine BACE-1), with 413 from 427 residues being identical. Wording on p 13 has been updated, and an additional reference was added describing the efficacy of BACE inhibition on dog CSF A β .

7. P9: Dose-limiting non-specific CNS effects (including impaired mobility and tremor) occurred in dogs at > 30 mg/kg/day and in rats at > 500 mg/kg/day. The dose limit in dogs seems relatively low. Did the dogs recover? Is this an acceptable risk? Have any humans shown such side effects in the clinical trials to date?

Answer: To translate the No Adverse Effect Level in the dog at 30 mg/kg/d to the human situation, we compared the compound exposure at this dose in the dog with the compound exposure at the highest dose used in the 3-months human study (85 mg/d) (p 24, Discussion section). The calculated safety margin of 11-fold is generally acceptable and in fact comfortable. The doses that will be used in the Phase III clinical trials will be lower than 85 mg/d (15 and 50 mg/d, manuscript in

preparation) and safety margins will be larger. Signs observed in dogs disappeared during the recovery period. Signs similar to the ones observed in dogs have not been seen in humans so far (see p 16 on the safety/tolerability in humans).

8. Do APP23 mice have CAA at 12-14 mo of age (the start of treatment in one study)? Did CNP520 reduce (or increase) CAA in this study? Was CNP520 treatment associated with any increase in CMH?

Answer: We performed a separate study of effects of chronic CNP520 treatment on CAA pathology in APP23 mice. We hope that the editor and reviewers agree with our plan to publish the results of this study in a separate communication.

9. It is interesting that there was a differential effect of CNP520 on microglia and astrocytes in the APP23 mice orally treated (in food) from ~13 mo to ~19 mo of age. Insoluble A β , plaques and sAPP β levels were reduced while sAPP α was increased, confirming a lowering of amyloidogenic processing of APP. There was a reduction of plaque-associated microglia (Iba-1, which is a microglia and macrophage marker), but no change in microglia distant from plaques. However, astrocytes both near and away from plaques were reduced. Does CNP520 have any direct effects on astrocytes (or microglia)? A further discussion of this interesting difference is warranted.

Answer: A formal investigation of CNP520 effects on microglia or astrocytes (in culture, or in wild-type animals) has not been performed. To better understand why CNP520 treatment had no effect on non-plaque-associated Iba1 positive microglia, we analyzed the correlation between normalized plaque area and Iba1 positive area, and added four new correlation graphs to EV Fig. 2. The graph EV Fig. 2D shows that there is a wide-spread activation of non-plaque-associated microglia, largely independent from plaque load, possibly suggesting some uncoupling of Iba1 positivity from deposited A β in the mice at this advanced stage of amyloidosis. In total numbers, the non-plaque-associated microglia encompasses the majority of microglia, while the plaque-associated (and treatment-responsive microglia) are a relatively small number. Therefore we do not observe a significant treatment response when analyzing total microglia. GFAP staining however, for yet unknown reasons, shows a correlation with plaque area for both plaque-associated and non-plaque-associated astrocytes, and a consistent treatment effect. It will be analyzed whether or not CAA of the small capillaries is linked to the astrocyte activation in the plaque-free areas (ongoing). The text on p 15 was expanded to better point to the different glia cell sub-populations.

10. P16: One subject on 86 mg CNP520 dropped out of the study due to "an AE of global amnesia (moderate), possibly due to transient ischemic attacks, according to a neurologist's examination". Was this AD deemed drug-related? Did this person, or any other subject experience any problems with mobility or other CNS effects?

Answer: The effect was possibly drug-related, but no other neurological symptoms have been observed. Wording on p 16 was extended to describe this better.

11. In Figure 8F, it appears possible that in plaque-bearing individuals (with a low CSF A β 42/40 ratio), there might be a possible effect of APOE4 genotype. Why were only 2 APOE4 carriers included in the 3 highest dose groups?

Answer: The 3 month treatment study was a dose-finding study in the "normal" elderly population, subjects were not enriched for presence of the APOE4 allele, and consequently, we found the expected frequency of APOE4 carriers (33/108, 30.5%). APOE4 genotype was not considered when randomizing participants to the dose groups and distribution was completely by chance. An additional sentence was added on p 20 to clarify this.

12. P22-23: Many of the phenotypes in BACE-1 KO mice are considered developmental. Do BACE-1 inducible conditional KO mice exist? If so, is BACE-1 KO post-development associated with any of these deleterious phenotypes (e.g. myelin and retinal changes)? [These animals may or may not exist but would be useful here.]

Answer: The group of Bob Vassar at Northwestern University, Chicago, IL is working on the characterization of tamoxifen-inducible conditional BACE-1 knockout mice. Preliminary results

have been presented at conferences, showing that the myelin and retina phenotypes observed in “full” genetic knock out mice are absent in the conditional knockout. This seems to support our interpretation of the developmental origin of the knockout phenotypes. However, these results are not yet published, and can therefore not be referenced here.

13. The authors should be sure to update their discussion regarding BACE-1 inhibition based upon any recent clinical trial announcements.

Answer: The recent announcements regarding verubecestat and lanabecestat (stopped at interim analysis for futility in trials of prodromal/early AD) and atabecestat (stopped due to liver enzyme elevation) have been added to the text. However, with only the company announcements, but no data in our hands, a serious discussion and comparison to CNP520 (in particular about the safety and tolerability profile) is not possible and was not done. Furthermore, the recently published paper on the verubecestat trial in mild-to moderate AD (Egan et al, 2018) has been added and is discussed when comparing the therapeutic vs preventive treatment approaches on p 22.

14. P28: "Methods employed in PK/PD studies in dogs and rats are described in Neumann (Neumann et al, 2015b)." Remove the first "Neumann"?

Answer: “Neumann” has been removed, sentence on p 29 now reads: ...in rats and dogs as described (Neumann et al, 2015).

15. Lastly, have the authors detected any gender-specific effects of CNP520 in any of their 5 human clinical trials thus far? Is there any reason to believe that the drug's efficacy or side effects might be different in men and women?

Answer: There were no apparent gender effects observed in these rather small clinical studies. Information is limited as of today; we need to await the outcome of the regular safety monitoring performed at the Generation studies. Preclinical toxicology studies also do not show gender differences, beyond of what is known to be related to the different metabolic activity of female vs male animals.

Reviewer 2

I have only a few minor questions, the answers to which I suggest the authors may consider, because they may be informative for the reader and add depth and more nuance to the Discussion. Foremost, it would be highly informative to put CNP520 into a more extensive context with the other BACE1 inhibitors in clinical trials. Some questions that come to mind are the following. How do the structure and drug properties of CNP520 compare to those of the other major inhibitors? Perhaps a comparative table or diagram would be useful. From what has been published for the major inhibitors, how do their adverse events (AEs) compare to those of CNP520? For CNP520 and the other inhibitors, are the AEs likely to be on-target or off-target? If off-target, could the differences in structures between the compounds be responsible for the off-target AEs? Comparison of CNP520 to verubecestat may be most informative in regard to these questions, as more data has been published on the latter than the other inhibitors. In particular, the authors should comment on these questions in relation to the recently published Egan et al (2018) New England J. of Medicine 378;18:1691 article on the EPOCH verubecestat trial results. Also, it would be informative for the authors to comment on the recently announced failure of JNJ-54861911 due to liver toxicity.

Answer: We do fully understand the reviewer’s wish for a more extensive comparison of CNP520 with other clinical compounds. We are, however, also dependent on the information published in peer-reviewed journals. In particular for the aspects of adverse events in longer clinical trials, there is some information available for verubecestat, and very little for atabecestat and lanabecestat (not more than the company announcement). The new data on verubecestat in mild-to moderate AD from Egan et al., 2018 have been incorporated in the manuscript (p 4 & 22), and have now been discussed in the context of disease state vs treatment chances for BACE-1 inhibitors. We have also now mentioned the tolerability data from Egan et al (p 25), but in the absence of more information, further discussion would be purely speculative. With longer term tolerability data from CNP520 and/or comparable data from verubecestat in prodromal AD and lanabecestat in early AD in the future, we will be hopefully in a better position to dissect the profile for the various compounds into

BACE-1 inhibitor class effects (possibly on-target side effects), effects related to exposure, metabolism, and selectivity differences.

Reviewer 3

1. Recent results of clinical trials of BACE1 inhibitors

Authors should include the reports regarding the trial of verubecestat (Egan et al., NEJM 2018) and lanabecestat (Alzforum or related website), and discuss about efficacy and the effect on model animals (reduction in CSF Abeta and Abeta deposition). Especially, result of verubecestat suggested that, in contrast to amyloid plaques in the brains of rodent AD model (as shown in this manuscript and the other compounds), decreased production of monomer Abeta did not lead to effective remodeling/clearance of senile amyloid developed in human brain. Please provide possible different characters at molecular level between rodents and humans and discuss this issue in appropriate manner.

Answer: We have extended the discussion on p 22 with the new verubecestat data. We interpret these data as a strong hint that at least in the symptomatic disease stages with an A β PET signal close to plateau, blocking of the generation of new A β does not halt the clinical course of the disease. Data from APP transgenic mice currently do not support the assumption that treatment with a BACE inhibitor does lead to clearance of pre-existing senile plaques, at least not within few months' observation period possible in mouse models. The reduction described is always a reduction relative to non-treated control mice but not a reduction compared to baseline. The discussion on p 26 has been extended to make clear that the animal and the human results are not in contradiction (significant net plaque removal not observed in both cases).

2. Effect of CNP520 on hair pigmentation

Results clearly suggested that the skin concentration of the compound significantly contributed the complete absence of hair depigmentation. If available, authors should show the data of PMEL17 processing in the skin of CNP520-treated mouse. Also, please provide the concentration of CNP520 in other model (i.e., dogs) as well as humans to strengthen the idea that distribution of CNP520 is a crucial factor.

Answer: Data on PMEL 17 processing in the mouse skin are not available. No skin tissue has been sampled from dogs or humans, and CNP520 concentrations in these species are not known. This information has been added to the discussion on p 23. Since we did not observe any hypopigmentation in 39 week dog studies, even at the highest dose, we concluded that there is a good safety window, and further investigations are neither required nor justified. Regarding the mechanism, the mouse data provide the only direct evidence supporting our hypothesis about the skin distribution of CNP520 being an important factor to explain the absence of hypopigmentation. However if we use CNP520 free plasma concentration in dogs and humans as a surrogate, the data on EV Fig 3 show that for most of the doses these concentrations are below the IC50 for the inhibition of BACE2, which points into the same direction. Nevertheless, routine dermatological assessments have been included in the study protocol for the ongoing Phase III studies, and this information has been added to the manuscript (p 23).

2nd Editorial Decision

21 August

Thank you for the submission of your revised manuscript to EMBO Molecular Medicine. We have now received the enclosed report from the referee who was asked to re-assess it. As you will see this reviewer is now supportive and I am happy to inform you that we will be able to accept your manuscript pending minor editorial amendments.

***** Reviewer's comments *****

Referee #2 (Comments on Novelty/Model System for Author):

The authors have done an excellent job revision the manuscript. It is now appropriate for publication.

Corresponding Author Name: Ulf neumann, Cristina Lopez Lopez

Manuscript Number: EMM-2018-09316